# Continuous Approximation of Momentum Methods with Explicit Discretization Error

## Abstract

Momentum-based optimization methods, such as Heavy-Ball (HB) and Nesterov's accelerated gradient (NAG), are essential in training modern deep neural networks. This work sheds light on the learning dynamics of momentum-based methods and how they behave differently than standard gradient descent (GD) in theory and practice. A promising approach to answer this question is investigating the continuous differential equations to approximate the discrete updates, an area requiring much attention for momentum methods. In this work, we take HB as a case study to investigate two important aspects of momentum methods. First, to enable a formal analysis of the Heavy-Ball momentum method, we propose a new continuous approximation, HB Flow (HBF), with a formulation that allows the control of discretization error to *arbitrary order*. As an application of HBF, we leverage it to investigate the implicit bias of HB by conducting a series of analyses on the diagonal linear networks to inspect the influence of momentum on the model's generalization property. We validate theoretical findings in numerical experiments, which confirm the significance of HBF as an effective proxy of momentum methods to bridge between discrete and continuous learning dynamics.

## 1 Introduction

Gradient descent (GD) and its variants momentum methods, such as Polyak's Heavy-Ball momentum (HB) [31], Nesterov's method of accelerated gradients (NAG) [27], and Adam [17], are at the core of the success of training deep neural networks. However, analyzing such discrete learning dynamics is challenging. Thus there is an extensive body of work in developing continuous approximations of GD and SGD (stochastic version of GD) [1; 2; 4; 5; 9; 14; 16; 22; 23; 24; 25]. These works advanced our understanding of the black-box generalization ability of the highly over-parameterized deep neural networks trained by GD and SGD. Despite the progress in analyzing GD, it remains unclear when and why momentum methods are effective. As momentum methods are now essential in many practical scenarios, such as the training of Transformer-based models [37], there is a critical need to understand the working mechanism of momentum methods.

Existing approaches for analyzing momentum methods typically use continuous differential equations to approximate discrete iterative updates. Representative works in the area include [19; 20; 33; 35; 40], which developed several second-order ODEs to analyze the convergence properties of HB and NAG. [18] demonstrated that the continuous approximation of HB and NAG, which solves the optimization problem $\min_\beta L(\beta)$, can be expressed as a rescaled gradient flow (RGF):

$$\dot{\beta} = -\frac{\nabla L(\beta)}{1 - \mu},\tag{1}$$

where $L$ is the objective function and $\mu$ is the momentum factor. Despite its convenience for studying discrete momentum methods, this approximation has two fundamental drawbacks. First, it overlooks a substantial portion of discretization error when approaching the continuous limit. Second, it is insufficient to differentiate between HB and GD due to its overly simplified nature. For example, the solutions of Eq. (1) and the gradient flow $\dot{\beta} = -\nabla L$ are similar, suggesting that GD and HB will converge to similar points. However, this is inconsistent with actual behavior (Fig. 1(a)), where GD and HB behave differently and converge to different points.

To address these issues, the recent work [11] adopted backward error analysis [4; 13] to construct a continuous equation for HB as a perturbed version of Eq. (1) that admits smaller discretization

error. While promising, these approaches still do not account for the gaps between actual discrete algorithms and their continuous approximations. Furthermore, these results are model-agnostic and do not consider other aspects of training, such as model architectures for specific learning tasks.

To this end, we develop HB Flow (HBF), a novel continuous approximation of the HB momentum method. HBF is designed to be *arbitrarily close* to HB (see Definition 2.1 for a formal definition). Specifically, we add a perturbation term to cancel the discretization error to the improved version of Eq. (1). This term helps to fill the gap between our continuous approximation and the discrete HB. We emphasize that one can control the precision of HBF as a proxy of the discrete learning dynamics of HB to arbitrary orders. Our approximation provides a more reliable foundation for continuously analyzing the less well-studied momentum methods.

In addition, we highlight that one immediate benefit of HBF is that we can apply it to analyze various intriguing properties of HB while the direct study of discrete learning dynamics is cumbersome. Among these properties, an important one is its implicit bias—the preference for particular solutions among all possible ones. For GD/GF, there are already abundant results for understanding their implicit bias for diagonal linear networks [3; 10; 29; 30; 41; 42], a non-convex setting that shares similar properties with more complex neural networks. As an interesting application of the proposed HBF, we analyze the implicit bias of HB for this popular model through the HBF to arbitrary orders of discretization error.

**Contributions.** Our primary contribution is a continuous approximation of HB, namely HBF, that can be arbitrarily close to the discrete learning dynamics of HB (Theorem 3.1). HBF provides a reliable foundation for analyzing HB in a continuous manner. We also present the comparison between HBF and continuous approximation of GD to different orders of approximation, as summarized in Table 1. In addition, as an application of HBF, we explicitly investigate the implicit bias of HBF for the diagonal linear network (Theorem 4.1 for arbitrary order of approximation and Corollary 4.2 for order 2) and reveal its difference compared to that of GF. Our findings are helpful for the understanding of the crucial while less studied implicit bias of momentum methods in a non-convex setting. This indicates the importance of the proposed HBF as a proxy of HB for analyzing its properties.

RELATED WORKS

The backward error analysis was applied to GD in [4] where the implicit gradient regularization effect was also proposed. With a similar idea, [25; 32] further developed continuous approximations of GD with discretization error that can be arbitrarily smaller. [33; 35; 39; 40] constructed continuous ODEs for momentum methods and primarily focused on the convergence aspects. [18; 11] studied the discretization error of continuous time limit of HB to the first 2 orders. [6] focused on the continuous limit of Adam. As a comparison, we firstly propose the HBF that can be $\mathcal{O}\left(\eta^{\alpha}\right)$-close to the discrete learning dynamics of HB.

The implicit bias of GD for various deep neural networks has been widely studied in, e.g., [16; 21; 34; 42] for linear networks and [7; 24; 26] for homogeneous networks. For diagonal linear networks, [3; 10; 29; 30; 41] examined the implicit bias of GD and revealed the interesting transition from kernel to rich regime by altering the scale of initialization. As a comparison, the study of implicit bias of momentum methods is not as fruitful as that of GD. [38; 12] investigated momentum methods for single-layer linear network and showed that they share a similar implicit bias with GD. [15] showed that momentum leads to better generalization in a special setting. The recent work [28] studied HB for diagonal linear networks using continuous approximation that is $\mathcal{O}\left(\eta\right)$-close to the discrete HB. To achieve a better understanding of the momentum methods for deep neural networks, in this paper, we characterize the implicit bias of HB for diagonal linear networks through the proposed HBF to *arbitrary order of discretization error*, which is absent in previous works, and compare it with that of GF to explicitly reveal their difference.

## 2 PRELIMINARIES

**Notations.** For a vector $\beta \in \mathbb{R}^d$ that depends on time $t$, we use $\dot{\beta}$ and $\ddot{\beta}$ to denote its first and second derivative with respect to time $t$, respectively. We use $\beta_j$ to denote its $j$-th component and $\|\beta\|_p$ for its $\ell_p$-norm. We use $\alpha \cdot \beta$ to denote the inner product and $\odot$ to denote elementwise product.

**Heavy-Ball momentum method.** HB [31] employs a two-step updating scheme [36], rather than the one-step manner of GD. Particularly, HB first accumulates the history of past iterations before updating the model parameter $\beta \in \mathbb{R}^d$, i.e., $m_{k+1} = \mu m_k - \eta \nabla L(\beta_k)$, $\beta_{k+1} = \beta_k + m_{k+1}$ where $\mu \in (0, 1)$ is the momentum factor, $\eta$ is the step size, $k$ is the iteration number, and $m \in \mathbb{R}^d$ is the momentum, which can be further written in a single equation

$$\beta_{k+1} = \beta_k - \eta \nabla L(\beta_k) + \mu \left( \beta_k - \beta_{k-1} \right). \tag{2}$$

To characterize the gap between the discrete learning dynamics of HB and its continuous approximation, we adopt the following definition.

**Definition 2.1** ($\mathcal{O}(\eta^\alpha)$-close continuous approximation of HB). *Let $\beta_k$ be the sequence given by Eq.* (2) *and $t_k = k\eta$. Given $\alpha \geq 1$, an ODE whose solution is $\beta(t)$ is $\mathcal{O}(\eta^\alpha)$-close continuous approximation of the discrete HB Eq.* (2) *if for a constant $C(T) > 0$ the supreme of the discretization error*

$$\sup_{0 \leq t_k \leq T} |\beta(t_k) - \beta_k| \leq C(T)\eta^\alpha.$$

## 3 CONTINUOUS APPROXIMATION OF MOMENTUM METHODS

In this section, we will propose a continuous differential equation that can be arbitrarily close to the discrete learning dynamics of HB. Our initial attempt is based on [18], where the authors showed that the $\mathcal{O}(\eta)$-close continuous approximation of HB is equivalent to a rescaled version of GF. To coincide with such observation and more precisely characterize the gap between HB and its continuous approximation, one may attempt to directly model HB by perturbing the RGF with an additional term that accounts for the discretization error using the backward error analysis [4]. However, this is problematic due to the fact that each iteration of momentum methods exploits the history of previous iterations, which renders the local error analysis unreliable since it ignores previous updates.

To address this difficulty and establish a more approachable $\mathcal{O}(\eta^\alpha)$-close continuous approximation of HB for any $\alpha \geq 1$, inspired by [25] which was originally designed for GF only locally , we propose a HB Flow (HBF)

$$\dot{\beta} = -G_k(\beta) - \eta \gamma_k(\beta), \text{ for } t \in [t_k, t_{k+1}) \tag{3}$$

to perform a global analysis instead of directly utilizing the backward error analysis in [4], which was previously discussed in [11]. In Eq. (3), $k$ denotes the iteration number for the corresponding discrete updates, $t_k = k\eta$, $G_k$ depends on $k$, and $\gamma_k$ accounts for the discretization error and also depends on $k$. The dependence on the iteration number $k$ of Eq. (3) indicates its piece-wise approximation nature, which will be made clearer in Theorem 3.1. Note that such dependence on the history of iterations also makes the analysis of the arbitrary order continuous approximation more challenging when compared to the case of GD, where each iteration only incorporates the information of the current gradient, i.e., $\dot{\beta} = -\nabla L(\beta) - \eta \gamma(\beta)$.

We now roughly outline the desired properties of a reliable HBF, then present the main results in Section 3, and defer the detailed technical proofs to Appendix A. Recall that in Definition 2.1 $\beta_k$ is the sequence given by HB (Eq. (2)) and $\beta(t)$ is the solution given by the HBF (Eq. (3)), we aim to find a $\gamma_k(\beta)$ that leads to a small discretization error $\varepsilon_k = \beta(t_k) - \beta_k$ to make the HBF a precise approximation of HB. For this purpose, considering the $\mathcal{O}(\eta)$ approximation of HB Eq. (1), we expect that for our HBF $G_k$ should have a form similar to $\nabla L(\beta)/(1 - \mu)$ while $\eta \gamma_k(\beta)$ should cancel the higher-order discretization error brought by the continuous approximation.

Specifically, for solution $\beta$ given by Eq. (3) and $t_k = k\eta$, Taylor expansion provides us

$$\beta(t_{k+1}) - \beta(t_k) = \eta \dot{\beta}(t_k^+) + \eta^2 I_k^+ = -\eta G_k - \eta^2 \gamma_k + \eta^2 I_k^+ \tag{4}$$

where $t_k^+$ means we approximate $t_k$ from $t > t_k$, $I_k^+ = \int_0^1 \ddot{\beta}\left(\eta(k + \tau)\right)(1 - \tau)d\tau$, and we apply Eq. (3) in the second equality. Similarly, $\beta(t_k) - \beta(t_{k-1}) = -\eta G_{k-1} - \eta^2 \gamma_{k-1} - \eta^2 I_k^-$. Combined with Eq. (4), we are able to give the relation between $\varepsilon_k$ and $\varepsilon_{k-1}$. Suppose that we can further find $\gamma_k$ such that $\varepsilon_k - \varepsilon_{k-1} = \mathcal{O}(\eta^\alpha)$ and $\varepsilon_k = \mathcal{O}(\eta^\alpha)$ by induction, then we can show that the HBF in Eq. (3) is $\mathcal{O}(\eta^\alpha)$-close to the discrete HB learning dynamics.

### 3.1 HBF with Arbitrary Order Closeness to HB

It is now left for us to find the desired $\gamma_k \in \mathbb{R}^d$ and $G_k \in \mathbb{R}^d$, which should consider the history of previous iterations and can degenerate into RGF, and reveal that the obtained HBF by doing so is indeed an $\mathcal{O}(\eta^\alpha)$ continuous approximation of HB. Below we formalize the aforementioned discussion and give the detailed expressions of $G_k$ and $\gamma_k$ in a recursive manner.

**Theorem 3.1** (HBF with $\mathcal{O}(\eta^\alpha)$ closeness to HB). *Let $k$ be the iteration number, $\eta$ be the step size, and $t_k = k\eta$, then the discretization error between the discrete HB momentum method Eq. (2) and the piece-wise HBF*

$$\dot{\beta} = -G_k(\beta) - \eta\gamma_k(\beta), \text{ for } t \in [t_k, t_{k+1})$$

*is $\mathcal{O}(\eta^\alpha)$ for $\alpha \geq 1$, i.e., HBF is $\mathcal{O}(\eta^\alpha)$-close continuous approximation of HB according to Definition 2.1, when*

$$G_k = \mu G_{k-1} + \nabla L, \ \gamma_k = \sum_{\sigma=0}^{\alpha-2} \eta^\sigma \gamma_k^{(\sigma)}, \tag{5}$$

*where the construction of $G_k$ intuitively resembles the discrete learning dynamics of HB[1] and we define the following notations to make the above expressions more concise:*

$$\gamma_k^{(\sigma)} = \sum_{j=0}^{k} \mu^{k-j} \chi_j^{(\sigma)}, \ \gamma_k^{(-1)} = G_k, \ \mathbf{L}_\beta^{(k,\sigma)} = \gamma_k^{(\sigma-1)} \cdot \nabla, \ \chi_0^{(0)} = \frac{(1+\mu)}{2} G_0 \nabla G_0$$

$$\mathcal{S}_{m,\sigma} = \left\{ (\sigma_1, \ldots, \sigma_m) \mid \sum_{i=1}^{m} \sigma_i = \sigma - m + 2, \forall i : \sigma_i \in \mathbb{Z}^+ \right\} \tag{6}$$

$$\chi_j^{(\sigma)} = \sum_{m=2}^{\sigma+2} \sum_{(\sigma_1, \ldots, \sigma_m) \in \mathcal{S}_{m,\sigma}} \frac{1}{m!} \Big[ (-1)^m \mathbf{L}_\beta^{(j,\sigma_1)} \cdots \mathbf{L}_\beta^{(j,\sigma_{m-1})} \gamma_j^{(\sigma_m-1)}$$

$$+ \mu \mathbf{L}_\beta^{(j-1,\sigma_1)} \cdots \mathbf{L}_\beta^{(j-1,\sigma_{m-1})} \gamma_{j-1}^{(\sigma_m-1)} \Big].$$

**Comparison with GD.** The gradient appeared in the continuous approximation of GD [25] is replaced by $G_k$ in our HBF, which depends on the iteration number $k$ and can be further simplified as

$$G_k = \frac{1 - \mu^{k+1}}{1 - \mu} \nabla L. \tag{7}$$

This difference is because each iteration of HB depends on the history of previous iterations. Such dependence is also reflected in the form of $\gamma_k^{(\sigma)}$: it incorporates information of all previous $\chi_j^{(\sigma)}$ with $j \leq k$ as shown in Eq. (6). By letting $\mu = 0$, all the dependence on $k$ will disappear and our results can recover those of GD. Interestingly, it is worth to mention that the difference between HBF and the continuous approximations of GD is closely related to the powers of $\eta(1+\mu)/(1-\mu)^2$ as we will show in Section 3.2.

**Arbitrary order continuous approximation of HB.** We note that $G_k \approx \nabla L/(1-\mu)$ for large $k$, which is consistent with the RGF in the $\mathcal{O}(\eta)$ continuous approximation of HB. Aside from this, we emphasize that HBF is more than just a rescaled version of GF—the differences are hidden in the higher order terms. Our results generalize the $\mathcal{O}(\eta)$ continuous approximation Eq. (1) and $\mathcal{O}(\eta^2)$ approximation of HB. In this sense, Theorem 3.1 provides a more reliable foundation for analyzing the rather less well-studied momentum methods through a continuous learning dynamics—it precisely indicates the extent of discrepancy between the results of HBF and the discrete ones. Particularly, to obtain a HBF that is $\mathcal{O}(\eta^\alpha)$-close to HB, one only needs to truncate the series of $\gamma_k$ to the order of $\alpha - 2$. As an important application, in Section 4, we will apply Theorem 3.1 to precisely characterize the implicit bias of momentum methods for diagonal linear network, a popular deep learning model that exhibits many insightful phenomena, such as the transition from kernel regime to rich regime that is common in more complex architectures.

| $\varepsilon_k = \mathcal{O}\left(\eta^\alpha\right)$ | GD | HB |
|---|---|---|
| $\alpha = 1$ | $\dot{\beta} = -\nabla L$ | $\dot{\beta} = -\frac{\nabla L}{(1-\mu)}$ [18] |
| $\alpha = 2$ | $\dot{\beta} = -\nabla L - \eta\frac{\nabla L \cdot \nabla^2 L}{2}$ [4] | $\dot{\beta} = -\frac{\nabla L}{1-\mu} - \eta\frac{1+\mu}{(1-\mu)^3}\frac{\nabla L \cdot \nabla^2 L}{2}$ Eq. (9) and [11] |
| $\alpha = 3$ | $\dot{\beta} = -\nabla L - \eta\frac{\nabla L \cdot \nabla^2 L}{2}$ $-\eta^2\left[\frac{\omega_1}{4} + \frac{\omega_2}{12}\right]$ [25; 32] | $\dot{\beta} = -\frac{\nabla L}{1-\mu} - \eta\frac{1+\mu}{(1-\mu)^3}\frac{\nabla L \cdot \nabla^2 L}{2}$ $-\frac{\eta^2(1+\mu)^2}{(1-\mu)^5}\left[\frac{\omega_1}{4} + \frac{(1+10\mu+\mu^2)\omega_2}{12(1+\mu)^2}\right]$ Eq. (10) |
| Arbitrary $\alpha$ | [25; 32] | Theorem 3.1 of this work |
| Discrete | $\beta_{k+1} = \beta_k - \nabla L(\beta_k)$ | $\beta_{k+1} = \beta_k - \eta\nabla L(\beta_k) + \mu\left(\beta_k - \beta_{k-1}\right)$ |

Table 1: Continuous approximations for GD and HB up to different orders of discretization error.

## 3.2 $\mathcal{O}\left(\eta^\alpha\right)$-CLOSE HBF FOR $\alpha = 2, 3$

There are basically three steps for finding a HBF that is $\mathcal{O}\left(\eta^\alpha\right)$-close to HB: **(1).** truncate $\gamma_k$ to the desired order $\alpha$, i.e., $\gamma_k = \sum_{\sigma=0}^{\alpha-2}\gamma_k^{(\sigma)}$; **(2).** from the smallest $\sigma = 0$ to $\sigma = \alpha - 2$, find all $\chi_j^{(\sigma)}$ with $j \leq k$ by identifying the corresponding $\mathcal{S}_{m,\sigma}$ with $m = \{2, \ldots, \sigma + 2\}$ for each $\sigma$; **(3).** derive the expression of $\gamma_k^{(\sigma)}$ for all $\sigma \leq \alpha - 2$ in a recursive manner using the relation $\gamma_k^{(\sigma)} = \sum_{j=0}^k \mu^{k-j}\chi_j^{(\sigma)}$.

Following this approach, one can find HBF that is $\mathcal{O}\left(\eta^\alpha\right)$-close to HB for arbitrary $\alpha$. In this section, we give the HBF with discretization error to the first several orders, i.e., $\alpha = 2, 3$. We also summarize these results in Table 1. Note that the case for $\alpha = 1$ states that HBF is a RGF, i.e., $\dot{\beta} = -\nabla L/(1-\mu)$, which might not fully characterize the difference between momentum methods and vanilla GD.

**HBF for $\alpha = 2$.** According to Theorem 3.1, there is only one term in the series of $\gamma_k$, i.e., $\gamma_k^{(0)}$. Recall that $\mathbf{L}_\beta^{k,0} = G_k \cdot \nabla$ and there is only one element in the set $\mathcal{S}_{m=2,\sigma=0}$, i.e., $\mathcal{S}_{m=2,\sigma=0} = \{(\sigma_1 = 0, \sigma_2 = 0)\}$, we obtain for $j \geq 1 : \chi_j^{(0)} = \frac{1}{2}\left[\mathbf{L}_\beta^{j,0}\gamma_j^{(-1)} + \mu\mathbf{L}_\beta^{j-1,0}\gamma_{j-1}^{(-1)}\right]$. Thus, using the definition of $\mathbf{L}_\beta^{j,0}$ and $\gamma_j^{(1)}$ in Theorem 3.1, we can immediately derive that

$$\gamma_k = \gamma_k^{(0)} = \frac{\nabla L \cdot \nabla^2 L}{2(1-\mu)^2}\sum_{j=0}^k \mu^{k-j}\left[(1 - \mu^{j+1})^2 + \mu(1 - \mu^j)^2\right]. \tag{8}$$

Typically the iteration number $k$ is large, therefore we further simplify the form of $\gamma_k$ for large iteration number $k$: $\gamma_k \approx \frac{1+\mu}{(1-\mu)^3}\frac{\nabla L \cdot \nabla^2 L}{2}$. This gives us HBF that is $\mathcal{O}\left(\eta^2\right)$-close to HB:

$$\dot{\beta} = -\frac{\nabla L}{1-\mu} - \eta\frac{1+\mu}{(1-\mu)^3}\frac{\nabla L \cdot \nabla^2 L}{2}, \tag{9}$$

which is consistent with the $\mathcal{O}\left(\eta^2\right)$ continuous approximation of HB in [11] while our derivation of HBF is in a different approach. It is worth to mention that when $\mu = 0$, HBF recovers the $\mathcal{O}\left(\eta^2\right)$ continuous approximation of GD as expected.

**HBF for $\alpha = 3$.** In this case we first truncate $\gamma_k$ to the order $\sigma = 1$, i.e., $\gamma_k = \gamma_k^{(0)} + \eta\gamma_k^{(1)}$. Since we already have $\chi_j^{(0)}$ in Eq. (8), we only need to find $\chi_j^{(1)}$ and $\gamma_k^{(1)}$, which can be done by first finding the collection of sets $\mathcal{S}_{m,\sigma}$ for $m = \{2, 3\}$ and $\sigma = 1$: $\mathcal{S}_{2,1} = \{(\sigma_1 = 1, \sigma_2 = 0), (\sigma_1 = 0, \sigma_2 = 1)\}, \mathcal{S}_{3,1} = \{(\sigma_1 = 0, \sigma_2 = 0, \sigma_3 = 0)\}$. We defer the detailed calculation to Appendix A

---

[1] $p_k = \mu p_{k-1} - \nabla L$.

and directly present the results here: $\gamma_k^{(1)} = \sum_{j=0}^k \mu^{k-j}(\Psi_j^{(1)} + \mu\Theta_j^{(1)})$ where, for convenience, we let

$$\Psi_j^{(1)} = \frac{1}{2}\left(\gamma_j^{(0)} \cdot \nabla^2 L + \nabla L \cdot \nabla\gamma_j^{(0)}\right) - \frac{1}{6}\nabla L \cdot \nabla\left(G_j \cdot \nabla G_j\right)$$

$$\Theta_j^{(1)} = \frac{1}{2}\left[\left(\gamma_j^{(0)} + \gamma_{j-1}^{(0)}\right) \cdot \nabla G_{j-1} + G_{j-1} \cdot \nabla\left(\gamma_j^{(0)} + \gamma_{j-1}^{(0)}\right)\right]$$

$$- \frac{G_{j-1} \cdot \nabla\left(G_j \cdot \nabla G_j - G_{j-1} \cdot \nabla G_{j-1}\right)}{6}.$$

For large $k$, let $\omega_1 = \left(\nabla L \cdot \nabla^2 L\right) \cdot \nabla^2 L$ and $\omega_2 = \nabla L \cdot \nabla\left(\nabla L \cdot \nabla^2 L\right)$, we can further simplify the form of $\gamma_k^{(1)}$ as $\gamma_k^{(1)} = \frac{(1+\mu)^2}{(1-\mu)^5}\left[\frac{\omega_1}{4} + \frac{(1+10\mu+\mu^2)\omega_2}{12(1+\mu)^2}\right]$, which further gives us the HBF that is $\mathcal{O}\left(\eta^3\right)$-close to HB:

$$\dot{\beta} = -\frac{\nabla L}{1-\mu} - \eta\frac{1+\mu}{(1-\mu)^3}\frac{\nabla L \cdot \nabla^2 L}{2} - \eta^2\frac{(1+\mu)^2}{(1-\mu)^5}\left[\frac{\omega_1}{4} + \frac{(1+10\mu+\mu^2)\omega_2}{12(1+\mu)^2}\right] \qquad (10)$$

According to Eq. (9), the $\mathcal{O}\left(\eta^2\right)$ approximation shows that momentum induces a stronger implicit gradient regularization (IGR, [4]), i.e., $\gamma_{\text{HB}} = (1+\mu)/(1-\mu)^3\gamma_{\text{GD}}$ where $\gamma_{\text{HB}}$ is the implicit regularization of HB while $\gamma_{\text{GD}}$ is that of GD. For the $\mathcal{O}\left(\eta^3\right)$-close HBF we can conclude that the difference between HB and GD is more complicated since HBF will rely more on $\omega_2$ which primarily depends on $\nabla^3 L$ than the continuous approximation of GD.

## 4 IMPLICIT BIAS OF MOMENTUM METHODS THROUGH HBF

The HBF proposed in Theorem 3.1 provides a reliable mathematical tool for analyzing a wide variety of properties of HB. A crucial while less well-studied aspect is its implicit bias, which is closely related to the generalization ability. To demonstrate the significance of HBF and obtain a deeper understanding of HB, in this section, we will characterize the implicit bias of HBF and reveal how it is connected with other sources such as model architectures. In particular, we focus on HBF for the diagonal linear network, a special deep neural network which shares several interesting phenomena with more complex architectures. This makes our setting a non-convex one. We begin with the definition of diagonal linear networks and a brief introduction of the corresponding regression setting.

**The formulation of diagonal linear networks.** An $L$-layer diagonal linear network [41] with parameter $\mathbf{w} = (\mathbf{w}_1, \mathbf{w}_2, \ldots, \mathbf{w}_L)$ where $\mathbf{w}_l \in \mathbb{R}^d$ for any $l \in \{1, \ldots, L\}$ is equivalent to a linear predictor $f(x; \mathbf{w}) = x^T(\mathbf{w}_L \odot \mathbf{w}_{L-1} \odot \cdots \odot \mathbf{w}_1)$. The diagonal linear network is a popular proxy model of more complicated deep neural networks. In this section, we focus on the 2-layer case, which, according to [41], induces an equivalent parameterization of $\mathbf{w} = \mathbf{w}_+ \odot \mathbf{w}_+ - \mathbf{w}_- \odot \mathbf{w}_-$.

For our task, given a dataset $\{(x_i, y_i)\}_{i=1}^n$ with $n$ samples where $x_i \in \mathbb{R}^d$ and $y_i \in \mathbb{R}$, we assume that $n < d$ and consider the regression problem. The quadratic loss is used for the linear predictor $f(x; \mathbf{w}) = x^T\mathbf{w}$, i.e., the empirical loss is $L(\mathbf{w}_+, \mathbf{w}_-) = \sum_{i=1}^n(x_i^T\mathbf{w} - y_i)^2/(2n)$. We use $X \in \mathbb{R}^{n \times d}$ to represent the data matrix and let $y = (y_1, \ldots, y_n)^T \in \mathbb{R}^n$. In the rest of this section, we use the HBF obtained in Theorem 3.1 to investigate its implicit bias and compare it with that of GF, which is discussed below.

**Implicit bias of GF for diagonal linear networks.** For the $\mathcal{O}\left(\eta\right)$ continuous approximation of GD, i.e., GF, [3; 41] showed that, if the model parameter $\mathbf{w}$ of the aforementioned 2-layer diagonal linear network converges to the interpolation solution and the initialization is $\mathbf{w}(0) = \mathbf{w}_+(0) \odot \mathbf{w}_+(0) - \mathbf{w}_-(0) \odot \mathbf{w}_-(0)$ with $\mathbf{w}_+(0) = \mathbf{w}_-(0)$, then the limit point of $\mathbf{w}$ is equivalent to the solution of the constrained optimization problem $\mathbf{w}(\infty) = \arg\min_{\mathbf{w}} \Lambda^{\text{GF}}(\mathbf{w}; \kappa^{\text{GF}})$, $s.t.$ $X\mathbf{w} = y$ where, let $\kappa_j^{\text{GF}} = \mathbf{w}_{+;j}(0)\mathbf{w}_{-;j}(0)$, the potential function $\Lambda^{\text{GF}}(\mathbf{w}; \kappa^{\text{GF}}) = \sum_{j=1}^d \Lambda_j^{\text{GF}}(\mathbf{w}; \kappa^{\text{GF}})$ and

$$\Lambda_j^{\text{GF}}(\mathbf{w}; \kappa^{\text{GF}}) = \frac{1}{4}\left[\mathbf{w}_j \operatorname{arcsinh}\left(\frac{\mathbf{w}_j}{2\kappa_j^{\text{GF}}}\right) - \sqrt{4(\kappa_j^{\text{GF}})^2 + \mathbf{w}_j^2} + 2\kappa_j^{\text{GF}}\right]. \qquad (11)$$

Note that the scale of $\kappa^{\text{GF}}$ controls the transition from rich regime to kernel regime, i.e., $\Lambda^{\text{GF}}(\mathbf{w}; \kappa^{\text{GF}}) \to \|\mathbf{w}\|_1$ for small $\kappa^{\text{GF}}$ while it approximates $\ell_2$-norm for large $\kappa^{\text{GF}}$ [41].

## 4.1 IMPLICIT BIAS OF HBF FOR DIAGONAL LINEAR NETWORKS

According to Theorem 3.1, the learning dynamics of the diagonal linear networks $f(x; \mathbf{w})$ can be written as

$$\dot{\mathbf{w}}_+ = -\frac{\nabla_{\mathbf{w}_+} L}{1 - \mu} - \eta\gamma_k^{\mathbf{w}_+}, \quad \dot{\mathbf{w}}_- = -\frac{\nabla_{\mathbf{w}_-} L}{1 - \mu} - \eta\gamma_k^{\mathbf{w}_-} \tag{12}$$

where we use $\gamma_k^{\mathbf{w}_+} \in \mathbb{R}^d$ and $\gamma_k^{\mathbf{w}_-} \in \mathbb{R}^d$ to represent the error terms for HBF of $\mathbf{w}_+$ and $\mathbf{w}_-$, respectively. Compared to the $\mathcal{O}(\eta)$ continuous approximation HBF, i.e., the RGF Eq. (1), Eq. (12) has one extra term that accounts for the high-order discretization error. One can check that the implicit bias of $\mathbf{w}$ under the RGF is similar to that of GF by following the approach in, e.g., [3]. However, this is not the case for HBF that is $\mathcal{O}(\eta^\alpha)$-close to HB for $\alpha \geq 1$ due to the $\gamma_k$ terms, which will be examined in the following.

**Theorem 4.1** (Implicit bias of HBF for diagonal linear networks). *If the dynamics of diagonal linear network $f(x; \mathbf{w}) = x^T\mathbf{w}$ where $\mathbf{w} = \mathbf{w}_+ \odot \mathbf{w}_+ - \mathbf{w}_- \odot \mathbf{w}_-$ follows HBF defined in Theorem 3.1 and if $\mathbf{w}(\infty)$ converges to an interpolation solution, let $\kappa_j(t) = \mathbf{w}_{+;j}(0)\mathbf{w}_{-;j}(0)\exp(-\eta\epsilon_j(t))$ where $\epsilon_j(t) = \int_0^t ds \left(\gamma_{k;j}^{\mathbf{w}_+}(s)/\mathbf{w}_{+;j}(s) + \gamma_{k;j}^{\mathbf{w}_-}(s)/\mathbf{w}_{-;j}(s)\right)$ and $\mathbf{w}_{+;j}(0) = \mathbf{w}_{-;j}(0)$, then $\mathbf{w}(\infty)$ satisfies that*

$$\mathbf{w}(\infty) = \arg\min_{\mathbf{w}} \Lambda(\mathbf{w}; \kappa) \quad s.t. \ X\mathbf{w} = y, \tag{13}$$

*where $\Lambda(\mathbf{w}; \kappa) = \sum_{j=1}^d \Lambda_j(\mathbf{w}, \infty; \kappa)$ and*

$$\Lambda_j(\mathbf{w}, t; \kappa) = \Lambda_j^{\mathrm{GF}}(\mathbf{w}; \kappa(t)) + \mathbf{w}_j\varphi_j(t),$$

$$\varphi_j(t) = \frac{\eta}{4}\int_0^t ds \left(\frac{\gamma_{k;j}^{\mathbf{w}_+}(s)}{\mathbf{w}_{+;j}(s)} - \frac{\gamma_{k;j}^{\mathbf{w}_-}(s)}{\mathbf{w}_{-;j}(s)}\right). \tag{14}$$

Note that we assume $\mathbf{w}_+(s) \neq 0$ and $\mathbf{w}_-(s) \neq 0$ for the dynamics. An immediate application of Theorem 4.1 shows that the implicit bias of $\mathcal{O}(\eta)$-close continuous approximation of HBF Eq. (1), i.e., RGF, is the same as that of GF by setting all the $\gamma_k^{\mathbf{w}_\pm}$ as 0. In this sense, RGF is not sufficient for the purpose of revealing the difference between the implicit bias of HB and GD through the continuous approximations, which suggests that the higher-order HBF is a more discerning choice.

**Comparison with the implicit bias of GF.** Compared to the implicit bias of GF in Eq. (11), there are two differences brought by the high-order correction terms of HBF: **(1)**. the potential function $\Lambda_j^{\mathrm{GF}}(\mathbf{w}; \kappa^{\mathrm{GF}})$ for GF becomes $\Lambda_j^{\mathrm{GF}}(\mathbf{w}; \kappa(\infty))$ for HBF where $\kappa(\infty)$ is different from $\kappa^{\mathrm{GF}}$, meaning that HBF induces an effect equivalent to a rescaling of the initialization $\kappa_j^{\mathrm{GF}}$; **(2)**. $\Lambda_j(\mathbf{w}, \infty; \kappa)$ additionally depends on the product $\mathbf{w}_j\varphi_j(\infty)$. A similar term will appear in the potential function of GF $\Lambda^{\mathrm{GF}}$ if the initialization no longer satisfies $\mathbf{w}_+(0) = \mathbf{w}_-(0)^2$. In this sense, HBF also brings an effect that is equivalent to breaking the symmetry of the initialization.

**Implicit bias of higher order continuous approximations of GD.** Another byproduct of Theorem 4.1 is the implicit bias of higher-order continuous approximation of GD for diagonal linear networks, which can be obtained by setting all $\mu$ appeared in $\gamma_k^{\mathbf{w}_\pm}$ as 0. In this sense, compared to GF, the implicit bias of higher-order continuous approximation of GD will also induce an effect that is equivalent to the modification of the initialization. Such effect has also been verified in GD [10] (and, interestingly, SGF [29]) and further reveals the reliability of high-order continuous approximations.

## IMPLICIT BIAS OF HBF FOR DIAGONAL LINEAR NETWORKS WITH $\alpha = 2$

We now focus on a special case $\alpha = 2$ as an explicit example to investigate the implicit bias of HBF and compare it with that of GF.

---

[2]One can show that $\Lambda^{\mathrm{GF}}$ will also depend on $\mathbf{w}^T\mathbf{c}$ for some vector $\mathbf{c}$ if $\mathbf{w}_+(0) \neq \mathbf{w}_-(0)$ following a similar approach.

**Corollary 4.2** (Implicit bias of $\mathcal{O}\left(\eta^2\right)$-close HBF for diagonal linear networks). *Under conditions of Theorem 4.1, if the $\mathcal{O}\left(\eta^2\right)$-close HBF is used, then*

$$\Lambda(\mathbf{w}; \kappa) = \sum_{j=1}^d \Lambda_j^{\mathrm{GF}}(\mathbf{w}, \kappa) + \frac{\eta(1+\mu)}{4(1-\mu)^2}\mathbf{w}^T(\nabla_\mathbf{w} L(0)),$$

*where*

$$\Phi_j = 4\int_0^\infty ds(\partial_{\mathbf{w}_j}L)^2 > 0, \ \ \mathbf{q} \in \mathbb{R}^d \text{ with } \mathbf{q}_i = \sqrt{\mathbf{w}_i^2(\infty) + 4\kappa_i^2(0)} - 2\kappa_i(0)$$

*and* $\kappa_j = \kappa_j(0)\exp\left[\frac{\eta(1+\mu)}{(1-\mu)^2}\left(-(1-\mu)^{-1}\Phi_j + \left(X^TX\mathbf{q}\right)_j/n\right)\right].$

Corollary 4.2 suggests that $\kappa$, which controls the transition from kernel regime to rich regime, is no longer the initialization $\kappa(0)$. Considering terms in the exponent of $\kappa_j(\infty)$, when the first term (the integral of $(\partial_\mathbf{w}L)^2$) is much larger than the second one, we can conclude that $\kappa_j(\infty) \approx \kappa_j(0)\exp\left(-\frac{4\eta(1+\mu)}{(1-\mu)^3}\int_0^\infty ds(\partial_{\mathbf{w}_j}L)^2\right) < \kappa_j(0)$, i.e., the initialization is equivalently decreased. As a result, $\Lambda(\mathbf{w}; \kappa)$ is closer to the $\ell_1$-norm thus the solution $\mathbf{w}(\infty)$ will enjoy better sparsity.

**Implication for difference of implicit bias between HB and GD.** Setting $\mu = 0$ in Corollary 4.2 gives us the implicit bias of $\mathcal{O}\left(\eta^2\right)$-close continuous approximation of GD, i.e., IGR [4] Flow (IGRF). Both IGRF and HBF have the initialization rescaling effect. The difference between them is closely connected with the parameter $\eta(1+\mu)/(1-\mu)^2$ and the value of $\Phi/(1-\mu)$. And the discrepancy between the implicit bias of HB and that of GD will be more obvious for large value of $\mu$. These observations stand in contrast to the case for $\mathcal{O}\left(\eta\right)$-close RGF, which cannot distinguish the implicit bias of HB from that of GD. In particular, compared to GD, HB converges faster thus $\Phi$ tends to be smaller for HBF than for IGRF, suggesting that, if HB converges too fast, $\kappa$ would be larger for HB and solutions of GD would enjoy better generalization properties. More interestingly, considering the case where $\Phi \gg X^TX\mathbf{q}/n$, e.g., $\kappa(0) \gg \mathbf{w}(\infty)$ such that $\mathbf{q} \approx 0$, if HB does not converge much faster than GD such that the difference of $\Phi$ between HBF and IGRF is not too significant, then HB would enjoy better sparsity than GD since it has a coefficient $(1+\mu)/(1-\mu)^2 > 1$ for $\Phi$ that further strengthens the initialization mitigation effect. Therefore, Corollary 4.2 indicates a trade-off between convergence rate and generalization benefit of momentum, which cannot be captured by lower order continuous approximation RGF.

## 5 NUMERICAL EXPERIMENTS

In this section, we show numerical experiments to verify our theoretical claims. We first present a 2d example, and then focus on the implicit bias of HB for a two-layer diagonal linear networks.

**Learning Dynamics in a 2-d Model** We first explore a simple 2-d model $f(x; a_1, a_2) = a_1a_2x$ where $a_1, a_2 \in \mathbb{R}$ are the model parameters and $x, y \in \mathbb{R}$ is the training data. The loss function is $L = \left(f(x; a_1, a_2) - y\right)^2/2$. All parameters $a_1, a_2$ satisfying $a_1a_2x = y$ are global minima. To show that higher-order HBFs are better approximations of HB, we visualize trajectories for different learning dynamics, i.e., GD, HB, RGF, HBF with $\alpha = 2$, and HBF with $\alpha = 3$, in Fig. 1(a). The trajectory of HBF with $\alpha = 3$ is closer to that of HB than both RGF and HBF with $\alpha = 2$. Furthermore, Fig. 1(a) also reveals that RGF is more similar to GD and it cannot capture the discrete learning dynamics of HB well. We also plot the norm of discretization errors $\|\varepsilon_k\|^2$ for these continuous approximations during training in Fig. 1(b), where HBF with $\alpha = 3$ has the lowest discretization error after several steps. These results validate the reliability of HBF as a proxy of HB.

**Implicit Bias of HB for Diagonal Linear Networks** We now investigate the implicit bias of HB for 2-layer diagonal linear networks $f(x; \mathbf{w}) = \mathbf{w}^T x = (\mathbf{w}_+ \odot \mathbf{w}_+ - \mathbf{w}_- \odot \mathbf{w}_-)^T x$ for a dataset $\{(x_i, y_i)\}_{i=1}^n$ where $x \in \mathbb{R}^d, y \in \mathbb{R}$. The empirical loss is $L(\mathbf{w}_+, \mathbf{w}_-) = \sum_i (f(x_i; \mathbf{w}) - y_i)^2$. We let $n < d$ and denote the ground truth solution as $\mathbf{w}^*$ such that $\mathbf{w}^{*T}x = y$. We let $\mathbf{w}^*$ be sparse. For

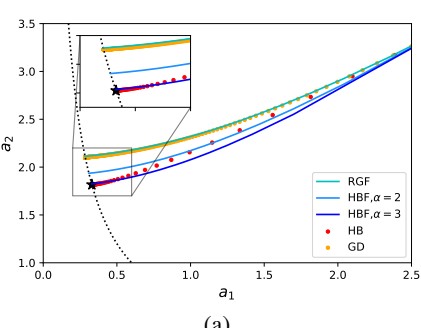 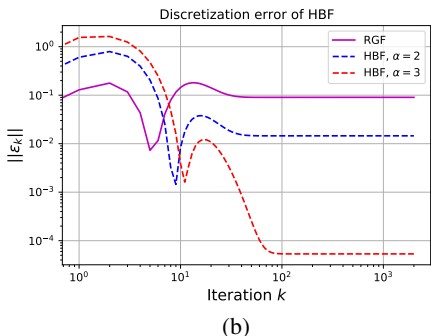

(a)                                            (b)

Figure 1: (a). Trajectories for learning dynamics of GD, HB, RGF, and $\mathcal{O}\left(\eta^2\right)$-close and $\mathcal{O}\left(\eta^2\right)$-close HBF in a 2-d model. All dynamics start from the same point ($a_1 = 2.8, a_2 = 3.5$). The convergence point of HB is denoted as a black star. The black dotted line denotes the set of all global minima. (b). Discretization errors for different continuous approximations of HB during training in (a).

a given scale $s$ we let $\kappa(0) = \mathbf{w}_+(0) \odot \mathbf{w}_-(0) = s^2(1, \ldots, 1)^T \in \mathbb{R}^d$. Our first experiment explores the discretization error, where we let $k$ denote the iteration count and first obtain $\mathbf{w}_k^{\text{HB}}$ by training $f(x; \mathbf{w})$ with HB. In addition, we also train $f(x; \mathbf{w})$ with RGF (Eq. (1)) and HBF (Corollary 4.2), respectively. We calculate the discretization error as $\|\mathbf{w}_k^{\text{HB}} - \mathbf{w}(t_k)\|_2^2$ for $\mathbf{w}(t_k)$ obtained from HBF or RGF and present the results in Fig. 2(a), where HBF enjoys smaller discretization error than RGF for different $\mu$, supporting our theoretical claims.

In our second experiment, we compare the implicit bias of HB with that of GD. Given $s$, we train $f(x; \mathbf{w})$ with GD and HB, respectively. We calculate the distance between the returned solution $\mathbf{w}(\infty)$ and the ground truth solution $\mathbf{w}^*$, i.e., $\|\mathbf{w}(\infty) - \mathbf{w}^*\|_2$, as a measure of generalization performance and report the results in Fig. 2(b). It can be seen that, when the initialization scale $s$ is small, solutions of GD generalize better than those of HB. This can be explained by Corollary 4.2: compared to GD, when $s$ is small, $L(\mathbf{w})$ decreases much faster for HB (green lines in Fig. 2(c)), which leads to a smaller $\int ds L(\mathbf{w})$ and weaker initialization mitigation effect, thus the solutions of HB generalize worse than GD solutions. Recall that in Corollary 4.2, as $\kappa_j(0) = s^2$ increases, $\Phi$ determines the generalization performances for HB and GD since it controls the extent of the initialization mitigation effect. Furthermore, $L(\mathbf{w})$ does not decrease much faster for HB than for GD (blue lines in Fig. 2(c)), thus GD and HB have a similar value of $\Phi$, which is further enhanced by a factor of $(1 + \mu)/(1 - \mu)^3$ for HB according to Corollary 4.2. As a result, HB solutions will generalize better than GD and the discrepancy between them is more significant for large $\mu$ (large $(1 + \mu)/(1 - \mu)^3$) as shown in Fig. 2(b).

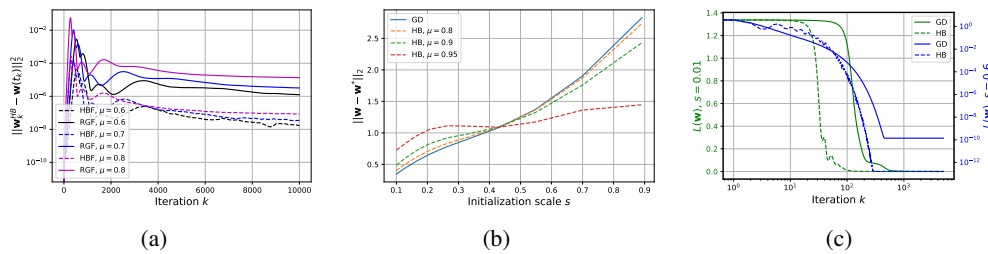

(a)                                    (b)                                    (c)

Figure 2: (a). Discretization errors $\|\mathbf{w}_k^{\text{HB}} - \mathbf{w}(t_k)\|_2^2$ for HBF (dotted lines) and RGF (solid lines), respectively, when training the 2-layer diagonal linear networks. (b). Generalization performances $\|\mathbf{w}(\infty) - \mathbf{w}^*\|_2$ for different initialization scales $s$ when $f(x; \mathbf{w})$ is trained by GD and HB with different values of $\mu$. (c). $L(\mathbf{w})$ during training processes of HB ($\mu = 0.9$) and GD for different $s$.

## 6 CONCLUSION

In this paper, we have established a new continuous approximation of HB, namely HBF, with $\mathcal{O}\left(\eta^{\alpha}\right)$ discretization error to the discrete HB for arbitrary $\alpha \geq 1$. Our results provide a reliable foundation for analyzing the less well-studied momentum methods through the continuous time limit. As an important and interesting application, we have studied the implicit bias of HBF for the popular proxy model diagonal linear networks and revealed the difference between the implicit bias of HB and that of GD which cannot be captured by RGF.

There are also many other momentum and adaptive algorithms, such as Adam [17] and AdaGrad [8], and more complex neural networks. A clear characterization of properties for these momentum methods and neural networks with methods developed in the current work is an interesting direction.

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

APPENDIX

In Appendix A, we provide proofs for Section 3. We present proofs of Section 4 in Appendix B. In Appendix C we show details of numerical experiments in Section 5 and 5.

## A  PROOFS FOR SECTION 3

We prove Theorem 3.1 in Appendix A.1 and give the details for the first several orders HBF in A.2.

### A.1  PROOF OF THEOREM 3.1

Recall that the discrete learning dynamics of HB is

$$\beta_{k+1} - \beta_k = -\eta \nabla L(\beta_k) + \mu(\beta_k - \beta_{k-1}), \tag{15}$$

where $\mu$ is the momentum factor and $k$ is the iteration number. Based on our discussion in Section 3, suppose that the continuous differential equation for HB is

$$\dot{\beta} = -G_k(\beta) - \eta \gamma_k(\beta) \tag{16}$$

for $t \in [t_k, t_{k+1})$ where $t_k = k\eta$ and the solution is $\beta(t)$, we expect that $\gamma_k$ could cancel higher-order discretization errors and $G_k$ would be closely related to rescaled gradient, i.e., $\nabla L/(1-\mu)$, when only the lowest order continuous approximation is considered, i.e., $\gamma_k$ and $G_k$ are chosen in such a way that $\beta(t_k)$ is close to $\beta_k$ in the sense that

$$\varepsilon_k = \beta(t_k) - \beta_k \tag{17}$$

is small. Inspired by [25], in the following, we will find such $\gamma_k$ and $G_k$ by investigating the properties of $\varepsilon_k$.

For this purpose, we begin with considering the Taylor expansion

$$\beta(t_{k+1}) - \beta(t_k) = \eta \dot{\beta}(t_k^+) + \eta^2 I_k^+ = -\eta G_k - \eta^2 \gamma_k + \eta^2 I_k^+ \tag{18}$$

where $t_k^+$ means we approximate $t_k$ from $t > t_k$, $I_k^+ = \int_0^1 \ddot{\beta}\left(\eta(k+s)\right)(1-s)ds$, and we use Eq. (16) in the second equality. Similarly,

$$\beta(t_k) - \beta(t_{k-1}) = -\eta G_{k-1} - \eta^2 \gamma_{k-1} - \eta^2 I_k^-.$$

We are now able to derive a recursive relation between $\varepsilon_{k+1}$ and $\varepsilon_k$ by subtracting Eq. (15) from both sides of Eq. (18):

$$\varepsilon_{k+1} - \varepsilon_k = -\eta \left[G_k(\beta(t_k)) - \nabla L(\beta_k)\right] - \eta^2 \gamma_k + \eta^2 I_k^+ - \mu(\beta_k - \beta_{k-1}). \tag{19}$$

Note that

$$\beta_k - \beta_{k-1} = \beta(t_k) - \beta(t_{k-1}) - (\varepsilon_k - \varepsilon_{k-1})$$
$$= -\eta G_{k-1} - \eta^2 \gamma_{k-1} - \eta^2 I_k^- - (\varepsilon_k - \varepsilon_{k-1}),$$

we obtain

$$\varepsilon_{k+1} - \varepsilon_k = \mu(\varepsilon_k - \varepsilon_{k-1}) - \eta\left[G_k - \mu G_{k-1} - \nabla L(\beta(t_k) - \varepsilon_k)\right]$$
$$+ \eta^2 \left[I_k^+ + \mu I_k^- - \gamma_k + \mu \gamma_{k-1}\right]. \tag{20}$$

To further establish the relation between $\varepsilon_k$ and $\varepsilon_{k-1}$, we present the following lemma and the proof can be found in A.1.1.

**Lemma A.1.** *For the continuous differential equation Eq. (16) and the discrete sequence given by Eq. (15), if $G_k(\beta(t_k)) = \mu G_{k-1}(\beta_{t_k}) + \nabla L(\beta(t_k))$ with $G_{-1} = 0$ and*

$$I_k^+ + \mu I_k^- - \gamma_k + \mu \gamma_{k-1} = \mathcal{O}\left(\eta^{\alpha-2}\right), \tag{21}$$

*then we have $\varepsilon_k = \mathcal{O}\left(\eta^\alpha\right)$ and*

$$\varepsilon_k - \varepsilon_{k-1} = \mathcal{O}\left(\eta^\alpha\right).$$

The above lemma ensures that the discretization error is of order $\mathcal{O}(\eta^{\alpha})$ as long as Eq. (21) is satisfied, which can be done by finding appropriate $\gamma_k$. For this purpose, we investigate the integral $I_k^{\pm}$ in the following lemma and present the proof in A.1.2.

**Lemma A.2.** *Under conditions of Lemma A.1, if we require that*

$$I_k^+ + \mu I_k^- = \gamma_k - \mu\gamma_{k-1}, \tag{22}$$

*then*

$$I_k^+ = \sum_{p=0}^{\infty}\sum_{q=2}^{p+2}\sum_{\sum_{j=1}^q \sigma_j = p-q+2}\frac{(-1)^q}{q!}\eta^p \mathbf{L}_{\beta}^{(k,\sigma_1)}\cdots\mathbf{L}_{\beta}^{(k,\sigma_{q-1})}\gamma_k^{(\sigma_q-1)} \tag{23}$$

$$I_k^- = \sum_{p=0}^{\infty}\sum_{q=2}^{p+2}\sum_{\sum_{j=1}^q \sigma_j = p-q+2}\frac{1}{q!}\eta^p \mathbf{L}_{\beta}^{(k-1,\sigma_1)}\cdots\mathbf{L}_{\beta}^{(k-1,\sigma_{q-1})}\gamma_{k-1}^{(\sigma_q-1)}. \tag{24}$$

Now suppose that $\gamma_k$ can be written as a series

$$\gamma_k = \sum_{\sigma=0}^{\infty}\eta^{\sigma}\gamma_k^{(\sigma)},\ \gamma_k^{(-1)} = G_k,$$

we can apply Lemma A.2 for any given $\sigma$ to solve Eq. (22) by requiring

$$\sum_{q=2}^{\sigma+2}\sum_{\sum_{j=1}^q \sigma_j = \sigma-q+2}\frac{\eta^{\sigma}}{q!}\left[(-1)^q\mathbf{L}_{\beta}^{(k,\sigma_1)}\cdots\mathbf{L}_{\beta}^{(k,\sigma_{q-1})}\gamma_k^{(\sigma_q-1)} + \mu\mathbf{L}_{\beta}^{(k-1,\sigma_1)}\cdots\mathbf{L}_{\beta}^{(k-1,\sigma_{q-1})}\gamma_{k-1}^{(\sigma_q-1)}\right]$$

$$= \eta^{\sigma}(\gamma_k^{(\sigma)} - \mu\gamma_{k-1}^{(\sigma)}). \tag{25}$$

Recall the definition in Theorem 3.1, we obtain that

$$\gamma_k^{(\sigma)} = \mu\gamma_{k-1}^{(\sigma)} + \chi_k^{(\sigma)} \implies \gamma_k^{(\sigma)} = \sum_{j=0}^{k}\mu^{k-j}\chi_j^{(\sigma)} \tag{26}$$

for $\chi_0^{(\sigma)} = \gamma_0^{(\sigma)}$. Now we can show that by truncating $\gamma_k$ to the order $\alpha-2$, i.e., $\gamma_k = \sum_{\sigma=0}^{\alpha-2}\eta^{\sigma}\gamma_k^{(\sigma)}$, the obtained HBF is an $\mathcal{O}(\eta^{\alpha})$-close continuous version of HB, since

$$\varepsilon_{k+1} - \varepsilon_k = \mathcal{O}(\eta^{\alpha}) + \mathcal{O}(\eta^{\alpha}) + \eta^2\left[I_k^+ + \mu I_k^- - \sum_{\sigma=0}^{\alpha-2}\eta^{\sigma}(\gamma_k^{(\sigma)} - \mu\gamma_{k-1}^{(\sigma)})\right]$$

$$= \eta^2\sum_{\sigma=\alpha-1}^{\infty}\eta^{\sigma}(\gamma_k^{(\sigma)} - \mu\gamma_{k-1}^{(\sigma)}) = \mathcal{O}(\eta^{\alpha}), \tag{27}$$

where we use Lemma A.1 in the first equality and Lemma A.2 in the second equality. We thus conclude that $\varepsilon_k = \mathcal{O}(\eta^{\alpha})$ according to Lemma A.1 again.

### A.1.1 PROOF FOR LEMMA A.1

*Proof.* By definition we have $\varepsilon_0 = 0 = \mathcal{O}(\eta^{\alpha})$ since the continuous differential equation and discrete HB start from the same point. Note that $G_{-1} = 0$ and $\gamma_{-1} = 0$ by definition. Recall that the first iterate of HB renders $\beta_1 = \beta_0 - \eta\nabla L(\beta_0)$ and that we let $G_0 = \nabla L(\beta_0)$, we have

$$\varepsilon_1 - \varepsilon_0 = \beta(t_1) - \beta_1 = \beta(t_0) + \eta\dot{\beta}(t_0) + \eta^2 I_0 - (\beta_0 - \eta\nabla L(\beta_0))$$

$$= \varepsilon_0 - \eta(G_0 - \nabla L(\beta_0)) - \eta^2\gamma_0 + \eta^2 I_0 = \mathcal{O}(\eta^{\alpha}). \tag{28}$$

The proof can now be completed by induction: suppose that for the iteration $k$

$$\varepsilon_k = \mathcal{O}(\eta^{\alpha}),\ \varepsilon_k - \varepsilon_{k-1} = \mathcal{O}(\eta^{\alpha}), \tag{29}$$

then, using Eq. (20) and the condition defined in Lemma A.1, we have for $k+1$

$$\varepsilon_{k+1} = \mathcal{O}(\eta^{\alpha}) - \eta\left[\nabla L(\beta(t_k)) - \nabla L(\beta(t_k) - \varepsilon_k)\right] + \mathcal{O}(\eta^{\alpha})$$

$$= \mathcal{O}(\eta^{\alpha}) - \eta\varepsilon_k \cdot \nabla^2 L(\beta(t_k)) = \mathcal{O}(\eta^{\alpha}). \tag{30}$$

$\square$

### A.1.2 PROOF FOR LEMMA A.2

*Proof.* We first rewrite $I_k^\pm$ as follows :

$$I_k^\pm = \frac{1}{\eta^2} \int_{k\eta}^{k\eta \pm \eta} \ddot\beta(\tau)(k\eta \pm \eta - \tau)d\tau$$

$$\overset{\tau' \leftarrow \tau - k\eta}{=} \frac{1}{\eta^2} \int_0^{\pm\eta} \left[ \sum_{n=0}^\infty \frac{1}{n!} \frac{d^n}{dt^n} \ddot\beta(k\eta)\tau'^n \right]^\pm (\pm\eta - \tau')d\tau'$$

$$= \sum_{n=0}^\infty \frac{(\pm\eta)^n}{(n+2)!} \frac{d^n}{dt^n} \ddot\beta(t_k^\pm) \tag{31}$$

where we use $\int_0^\eta \tau'^n(\eta - \tau')d\tau' = \frac{\eta^{n+2}}{n+1} - \frac{\eta^{n+2}}{n+2} = \frac{\eta^{n+2}}{(n+1)(n+2)}$ in the last equality. To continue, we need the expression of $d^n\beta/dt^n$ and we start with $t \to t_k^+$:

$$\frac{d^n}{dt^n}\beta(t_k^+) = \frac{d}{dt}\left( \frac{d^{n-1}}{dt^n}\beta(t_k^+) \right)$$

$$= \dot\beta(t_k^+) \cdot \nabla \left( \frac{d^{n-1}}{dt^n}\beta(t_k^+) \right)$$

$$= -(G_k + \eta\gamma_k) \cdot \nabla \left( \frac{d^{n-1}}{dt^n}\beta(t_k^+) \right)$$

$$= (-1)^n (\mathbf{L}_\beta^{(k)})^{n-1} (G_k + \eta\gamma_k) \tag{32}$$

where we denote the differential operator $\mathbf{L}_\beta^{(k)} = (G_k + \eta\gamma_k) \cdot \nabla$ and use Eq. (16) in the third equality. Now suppose that $\gamma_k$ can be written as a series

$$\gamma_k = \sum_{\sigma=0}^\infty \eta^\sigma \gamma_k^{(\sigma)}, \ \gamma_k^{(-1)} = G_k,$$

then Eq. (32) becomes

$$\frac{d^n}{dt^n}\beta(t) = (-1)^n \left( \sum_{\sigma_1=0}^\infty \eta^{\sigma_1}\gamma_k^{(\sigma_1-1)} \cdot \nabla \right) \cdots$$

$$\cdots \left( \sum_{\sigma_{n-1}=0}^\infty \eta^{\sigma_{n-1}}\gamma_k^{(\sigma_{n-1}-1)} \cdot \nabla \right) \left( \sum_{\sigma_{n-1}=0}^\infty \eta^{\sigma_n}\gamma_k^{(\sigma_n-1)} \right)$$

$$= (-1)^n \sum_{\sigma_1,\ldots,\sigma_n=0}^\infty \eta^{\sum_{j=1}^n \sigma_j} \mathbf{L}_\beta^{(k,\sigma_1)} \cdots \mathbf{L}_\beta^{(k,\sigma_{n-1})} \gamma_k^{(\sigma_n-1)} \tag{33}$$

Combined with Eq. (31), we obtain the form of $I_k^+$ for $t \in (t_k, t_{k+1})$ as

$$I_k^+ = \sum_{n=0}^\infty \frac{\eta^n}{(n+2)!} \frac{d^{n+2}}{dt^{n+2}}\beta(t)$$

$$= \sum_{n=0}^\infty \sum_{\sigma_1,\ldots,\sigma_{n+2}=0}^\infty \frac{(-1)^{n+2}}{(n+2)!} \eta^{n+\sum_{j=1}^{n+2}\sigma_j} \mathbf{L}_\beta^{(k,\sigma_1)} \cdots \mathbf{L}_\beta^{(k,\sigma_{n+1})} \gamma_k^{(\sigma_{n+2}-1)}$$

$$= \sum_{n=0}^\infty \sum_{m=0}^\infty \sum_{\sum_{j=1}^{n+2}\sigma_j=m} \frac{(-1)^{n+2}}{(n+2)!} \eta^{n+m} \mathbf{L}_\beta^{(k,\sigma_1)} \cdots \mathbf{L}_\beta^{(k,\sigma_{n+1})} \gamma_k^{(\sigma_{n+2}-1)}$$

$$= \sum_{p=0}^\infty \sum_{q=2}^{p+2} \sum_{\sum_{j=1}^q \sigma_j=p-q+2} \frac{(-1)^q}{q!} \eta^p \mathbf{L}_\beta^{(k,\sigma_1)} \cdots \mathbf{L}_\beta^{(k,\sigma_{q-1})} \gamma_k^{(\sigma_q-1)} \tag{34}$$

where we let $p \leftarrow n + \sum_{j=1}^{n+2} \sigma_j, q \leftarrow n + 2$, in the last equality. Similarly, when $t \rightarrow t_k^-$, we have

$$\frac{d^n}{dt^n} \beta(t_k^-) = (-1)^n (\mathbf{L}_\beta^{(k-1)})^{n-1} (G_{k-1} + \eta \gamma_{k-1})$$

which implies that

$$I_k^- = \sum_{p=0}^{\infty} \sum_{q=2}^{p+2} \sum_{\sum_{j=1}^q \sigma_j = p-q+2} \frac{1}{q!} \eta^p \mathbf{L}_\beta^{(k-1,\sigma_1)} \cdots \mathbf{L}_\beta^{(k-1,\sigma_{q-1})} \gamma_{k-1}^{(\sigma_q - 1)}. \tag{35}$$

$\square$

## A.2 $\mathcal{O}(\eta^\alpha)$-CLOSE HBF FOR A SPECIFIC $\alpha$

In this section, we derive the form of $\mathcal{O}(\alpha)$-close HBF for given a specific $\alpha$. There are basically three steps to find a HBF that is $\mathcal{O}(\eta^\alpha)$-close to HB:

1. truncate $\gamma_k$ to the desired order $\alpha$, i.e, $\gamma_k = \sum_{\sigma=0}^{\alpha-2} \gamma_k^{(\sigma)}$;

2. from the smallest $\sigma$, find all $\chi_j^{(\sigma)}$ with $j \leq k$ by finding the corresponding $\mathcal{S}_{m,\sigma}$ with $m = \{2, \ldots, \sigma + 2\}$ for each $\sigma$;

3. derive the expression of $\gamma_k^{(\sigma)}$ for all $\sigma \leq \alpha - 2$ in a recursive manner using the relation $\gamma_k^{(\sigma)} = \sum_{j=0}^k \mu^{k-j} \chi_j^{(\sigma)}$.

In the following, we give the cases for $\alpha = 2$ and $3$ as examples. With this approach, one can in fact find HBF with arbitrary order of closeness to HB.

### A.2.1 $\alpha = 2$.

According to Theorem 3.1, the series of $\gamma_k$ is truncated to the first term, i.e., $\gamma_k = \eta^0 \gamma_k^{(0)}$, where $\gamma_k = \sum_{j=0}^k \mu^{k-j} \chi_j^{(0)}$. Thus the first step is to find $\chi_j^{(0)}$, which can be given by first identifying the set $\mathcal{S}$:

$$\mathcal{S}_{m=2,\sigma=0} = \{(\sigma_1 = 0, \sigma_2 = 0)\}, \tag{36}$$

therefore there is only one term in $\chi_j^{(0)}$:

$$\chi_j^{(0)} = \frac{1}{2} \left[ \mathbf{L}_\beta^{j,0} \gamma_j^{(-1)} + \mu \mathbf{L}_\beta^{j-1,0} \gamma_{j-1}^{(-1)} \right].$$

Recall that

$$\gamma_j^{(-1)} = G_j = \frac{1 - \mu^{j+1}}{1 - \mu} \nabla L, \tag{37}$$

which, according to our definition in Theorem 3.1, leads to

$$\mathbf{L}_\beta^{j,0} = \gamma_j^{(-1)} \cdot \nabla = G_j \cdot \nabla,$$

we obtain that

$$\chi_j^{(0)} = \frac{1}{2} \left[ G_j \cdot \nabla G_j + \mu G_{j-1} \cdot \nabla G_{j-1} \right]$$

$$= \frac{1}{2(1-\mu)^2} \left[ (1 - \mu^{j+1})^2 + \mu(1 - \mu^j)^2 \right] \nabla L \cdot \nabla^2 L. \tag{38}$$

Thus

$$\gamma_k^{(0)} = \frac{1}{2} \sum_{j=0}^k \mu^{k-j} \left[ G_j \cdot \nabla G_j + \mu G_{j-1} \cdot \nabla G_{j-1} \right]$$

$$= \frac{\nabla L \cdot \nabla^2 L}{2(1-\mu)^2} \sum_{j=0}^k \mu^{k-j} \left[ (1 - \mu^{j+1})^2 + \mu(1 - \mu^j)^2 \right]$$

$$= \frac{\nabla L \cdot \nabla^2 L}{2(1-\mu)^2} \sum_{j=0}^k \left[ (1 + \mu)\mu^{k-j} + \mu^{k+1}(\mu^j(1 + \mu) - 4) \right]. \tag{39}$$

When $k$ is larege, the above expression can be simplified as

$$\gamma_k^{(0)} \approx \frac{(1+\mu)\sum_{j=0}^{k}\mu^j}{2(1-\mu)^2}\nabla L \cdot \nabla^2 L \approx \frac{1+\mu}{2(1-\mu)^3}\nabla L \cdot \nabla^2 L.$$

### A.2.2 $\alpha = 3$.

Similarly, in this case we first truncate the series of $\gamma_k$ to the desired order, i.e., $\gamma_k = \gamma_k^{(0)} + \eta\gamma_k^{(1)}$ where we have already obtained $\gamma_k^{(0)}$ in the last section, thus we only need to find $\gamma_k^{(1)}$ and $\chi_k^{(1)}$, which can be done by first finding the set $\mathcal{S}_{m=2,\sigma=1}$ and $\mathcal{S}_{m=3,\sigma=1}$:

$$\mathcal{S}_{2,1} = \{(\sigma_1 = 1, \sigma_2 = 0), (\sigma_1 = 0, \sigma_2 = 1)\},$$
$$\mathcal{S}_{3,1} = \{(\sigma_1 = 0, \sigma_2 = 0, \sigma_3 = 0)\}.$$

Therefore there are three terms of $\chi_j^{(1)}$:

$$\chi_j^{(1)} = \frac{1}{2}\left[\mathbf{L}_\beta^{j,1}\gamma_j^{(-1)} + \mu\mathbf{L}_\beta^{j-1,1}\gamma_{j-1}^{(-1)}\right] + \frac{1}{2}\left[\mathbf{L}_\beta^{j,0}\gamma_j^{(0)} + \mu\mathbf{L}_\beta^{j-1,0}\gamma_{j-1}^{(0)}\right]$$
$$- \frac{1}{6}\left[\mathbf{L}_\beta^{j,0}\mathbf{L}_\beta^{j,0}\gamma_j^{(-1)} - \mu\mathbf{L}_\beta^{j-1,0}\mathbf{L}_\beta^{j-1,0}\gamma_{j-1}^{(-1)}\right]. \tag{40}$$

Recall that $\gamma_j^{(-1)} = G_j$, $\mathbf{L}_\beta^{j,0} = G_j \cdot \nabla$, and $\mathbf{L}_\beta^{j,1} = \gamma_j^{(0)} \cdot \nabla$, the first line of Eq. (40) is

$$\frac{1}{2}\left[\gamma_j^{(0)} \cdot \nabla G_j + \mu\gamma_{j-1}^{(0)} \cdot \nabla G_{j-1} + G_j \cdot \nabla\gamma_j^{(0)} + \mu G_{j-1} \cdot \nabla\gamma_{j-1}^{(0)}\right] \tag{41}$$

while the second line is

$$-\frac{1}{6}\left[G_j \cdot \nabla\left(G_j \cdot \nabla G_j\right) - \mu G_{j-1} \cdot \nabla\left(G_{j-1} \cdot \nabla G_{j-1}\right)\right]. \tag{42}$$

To simplify these terms, we can either replace all $\gamma_j^{(0)}$ with the expression in Eq. (39) and write $G_j$ explicitly, or notice the recursive relation between $G_j$ and $G_{j-1}$ in Theorem 3.1, i.e. , $G_j = \mu G_{j-1} + \nabla L$, then Eq. (41) becomes

$$\frac{1}{2}\left[\gamma_j^{(0)} \cdot \nabla^2 L + \nabla L \cdot \nabla\gamma_j^{(0)}\right] + \frac{\mu}{2}\left[\left(\gamma_j^{(0)} + \gamma_{j-1}^{(0)}\right) \cdot \nabla G_{j-1} + G_{j-1} \cdot \nabla\left(\gamma_j^{(0)} + \gamma_{j-1}^{(0)}\right)\right]$$

and Eq. (42) is now

$$-\frac{1}{6}\nabla L \cdot \nabla\left(G_j \cdot \nabla G_j\right) - \frac{\mu}{6}G_{j-1} \cdot \nabla\left(G_j \cdot \nabla G_j - G_{j-1} \cdot \nabla G_{j-1}\right). \tag{43}$$

Summing over these terms gives us $\chi_j^{(1)}$:

$$\chi_j^{(1)} = \Psi_j^{(1)} + \mu\Theta_j^{(1)} \tag{44}$$

where

$$\Psi_j^{(1)} = \frac{1}{2}\left(\gamma_j^{(0)} \cdot \nabla^2 L + \nabla L \cdot \nabla\gamma_j^{(0)}\right) - \frac{1}{6}\nabla L \cdot \nabla\left(G_j \cdot \nabla G_j\right)$$
$$\Theta_j^{(1)} = \frac{1}{2}\left[\left(\gamma_j^{(0)} + \gamma_{j-1}^{(0)}\right) \cdot \nabla G_{j-1} + G_{j-1} \cdot \nabla\left(\gamma_j^{(0)} + \gamma_{j-1}^{(0)}\right)\right]$$
$$- \frac{1}{6}G_{j-1} \cdot \nabla\left(G_j \cdot \nabla G_j - G_{j-1} \cdot \nabla G_{j-1}\right).$$

We can now find $\gamma_k^{(1)}$ through its definition:

$$\gamma_k^{(1)} = \sum_{j=0}^{k}\mu^{k-j}\chi_j^{(1)} = \sum_{j=0}^{k}\mu^{k-j}\Psi_j^{(1)} + \mu\sum_{j=0}^{k}\mu^{k-j}\Theta_j^{(1)}. \tag{45}$$

In the following, we derive the form of $\gamma_k^{(1)}$ When $k$ is large. According to Eq. (39), we have

$$
\mu^{k-j}\gamma_j^{(0)} = \mu^{k-j}\frac{\nabla L \cdot \nabla^2 L}{2(1-\mu)^2}\sum_{i=0}^{j}\left[(1+\mu)\mu^{j-i}+\mu^{j+1}(\mu^i(1+\mu)-4)\right]
$$

$$
= \mu^{k-j}\frac{\nabla L \cdot \nabla^2 L}{2(1-\mu)^2}\left[\frac{(1+\mu)(1-\mu^{j+1})}{1-\mu}+\frac{\mu^{j+1}(1+\mu)(1-\mu^{j+1})}{1-\mu}-4(j+1)\mu^{j+1}\right]
$$

$$
= \frac{\nabla L \cdot \nabla^2 L}{2(1-\mu)^2}\left[\frac{(1+\mu)(\mu^{k-j}-\mu^{k+1})}{1-\mu}+\frac{\mu^{k+1}(1+\mu)(1-\mu^{j+1})}{1-\mu}-4(j+1)\mu^{k+1}\right]
$$

$$
= \frac{\nabla L \cdot \nabla^2 L}{2(1-\mu)^2}\left[\frac{(1+\mu)\mu^{k-j}}{1-\mu}-\frac{\mu^{k+j+1}(1+\mu)}{1-\mu}-4(j+1)\mu^{k+1}\right]
$$

$$
\approx \mu^{k-j}\frac{(1+\mu)}{2(1-\mu)^3}\nabla L \cdot \nabla^2 L \tag{46}
$$

and, according to Eq. (37),

$$
\mu^{k-j}G_j \cdot \nabla G_j = \frac{\mu^{k-j}(1-2\mu^{j+1}+\mu^{2(j+1)})}{(1-\mu)^2}\nabla L \cdot \nabla^2 L \approx \frac{\mu^{k-j}}{(1-\mu)^2}\nabla L \cdot \nabla^2 L. \tag{47}
$$

Combining Eq. (46) and (47) gives the form of $\mu^{k-j}\Psi_j^{(1)}$ when $k$ is large:

$$
\mu^{k-j}\Psi_j^{(1)} \approx \frac{\mu^{k-j}(1+\mu)}{4(1-\mu)^3}\left[(\nabla L \cdot \nabla^2 L)\cdot \nabla^2 L+\nabla L \cdot \nabla\left(\nabla L \cdot \nabla^2 L\right)\right]
$$

$$
-\frac{\mu^{k-j}}{6(1-\mu)^2}\nabla L \cdot \nabla\left(\nabla L \cdot \nabla^2 L\right)
$$

which immediately leads to

$$
\sum_{j=0}^{k}\mu^{k-j}\Psi_j^{(1)}
$$

$$
\approx \frac{(1+\mu)}{4(1-\mu)^4}\left[(\nabla L \cdot \nabla^2 L)\cdot \nabla^2 L+\nabla L \cdot \nabla\left(\nabla L \cdot \nabla^2 L\right)\right]-\frac{\nabla L \cdot \nabla\left(\nabla L \cdot \nabla^2 L\right)}{6(1-\mu)^3}
$$

$$
= \frac{1}{4(1-\mu)^4}\left[(1+\mu)(\nabla L \cdot \nabla^2 L)\cdot \nabla^2 L+\frac{(1+5\mu)}{3}\nabla L \cdot \nabla\left(\nabla L \cdot \nabla^2 L\right)\right]. \tag{48}
$$

The left part is now deriving the form of $\mu^{k-j}\Theta_j^{(1)}$, which can be done by first finding

$$
\mu^{k-j}\gamma_j^{(0)}\cdot \nabla G_{j-1} \approx \mu^{k-j}\frac{(1+\mu)}{2(1-\mu)^3}(\nabla L \cdot \nabla^2 L)\cdot \nabla G_{j-1}
$$

$$
\approx \mu^{k-j}\frac{(1+\mu)}{2(1-\mu)^4}(\nabla L \cdot \nabla^2 L)\cdot \nabla^2 L \approx \mu^{k-j}\gamma_{j-1}^{(0)}\cdot \nabla G_{j-1} \tag{49}
$$

and

$$
\mu^{k-j}G_{j-1}\cdot \nabla\left(G_j \cdot \nabla G_j-G_{j-1}\cdot \nabla G_{j-1}\right) \approx \frac{2\mu^{k-j}}{(1-\mu)^3}\nabla L \cdot \nabla\left(\nabla L \cdot \nabla^2 L\right), \tag{50}
$$

thus

$$
\sum_{j=0}^{k}\mu^{k-j}\Theta_j^{(1)} \approx \frac{(1+\mu)}{2(1-\mu)^5}\left[(\nabla L \cdot \nabla^2 L)\cdot \nabla^2 L+\nabla L \cdot \nabla\left(\nabla L \cdot \nabla^2 L\right)\right]. \tag{51}
$$

Combing this equation with Eq. (48), we can now conclude the form of $\gamma_k^{(1)}$ when $k$ is large:

$$\gamma_k^{(1)} = \sum_{j=0}^{k} \mu^{k-j} \left( \Psi_j^{(1)} + \mu \Theta_j^{(1)} \right)$$

$$\frac{1}{4(1-\mu)^4} \left[ (1+\mu)(\nabla L \cdot \nabla^2 L) \cdot \nabla^2 L + \frac{(1+5\mu)}{3} \nabla L \cdot \nabla \left( \nabla L \cdot \nabla^2 L \right) \right]$$

$$+ \frac{\mu(1+\mu)}{2(1-\mu)^5} \left[ (\nabla L \cdot \nabla^2 L) \cdot \nabla^2 L + \nabla L \cdot \nabla \left( \nabla L \cdot \nabla^2 L \right) \right]$$

$$= \frac{(1+\mu)^2}{4(1-\mu)^5} \left[ (\nabla L \cdot \nabla^2 L) \cdot \nabla^2 L + \frac{1 + 10\mu + \mu^2}{3(1+\mu)^2} \nabla L \cdot \nabla \left( \nabla L \cdot \nabla^2 L \right) \right] \tag{52}$$

Note that when $\mu = 0$ we recover the result of GD, i.e., $\gamma_k^{(1)} = \frac{(\nabla L \cdot \nabla^2 L) \cdot \nabla^2 L}{4} + \frac{\nabla L \cdot \nabla \left( \nabla L \cdot \nabla^2 L \right)}{12}$.

# B  PROOFS FOR SECTION 4

Given data $(x_i, y_i)$, the architecture of 2-layer diagonal linear network is

$$f(x_i; \mathbf{w}) = x_i^T (\mathbf{w}_+ \odot \mathbf{w}_+ - \mathbf{w}_- \odot \mathbf{w}_-) = \sum_{j=1}^{d} x_{i;j} \left( \mathbf{w}_{+;j}^2 - \mathbf{w}_{-;j}^2 \right) \tag{53}$$

and the empirical loss function is

$$L(\mathbf{w}) = \frac{1}{2n} \sum_{i=1}^{n} (f(x_i; \mathbf{w}) - y_i)^2.$$

We let $r = (r_1, \ldots, r_n)^T \in \mathbb{R}^n$ be the residual where $\forall i : r_i = f(x_i; \mathbf{w}) - y_i$. According to Theorem 3.1, the HBF learning dynamics of model parameters $\mathbf{w}_+$ and $\mathbf{w}_-$ will be

$$\dot{\mathbf{w}}_+ = -\frac{\nabla_{\mathbf{w}_+} L}{1 - \mu} - \eta \gamma_k^{\mathbf{w}_+}, \quad \dot{\mathbf{w}}_- = -\frac{\nabla_{\mathbf{w}_-} L}{1 - \mu} - \eta \gamma_k^{\mathbf{w}_-} \tag{54}$$

where we use $\gamma_k^{\mathbf{w}_+} \in \mathbb{R}^d$ and $\gamma_k^{\mathbf{w}_-} \in \mathbb{R}^d$ to represent the error terms for HBF of $\mathbf{w}_+$ and $\mathbf{w}_-$, respectively, and the gradients are

$$\nabla_{\mathbf{w}} L = \frac{1}{n} X^T r, \tag{55}$$

$$\nabla_{\mathbf{w}_+} L = 2\mathbf{w}_+ \odot \nabla_{\mathbf{w}} L, \quad \nabla_{\mathbf{w}_-} L = -2\mathbf{w}_- \odot \nabla_{\mathbf{w}} L. \tag{56}$$

Using the expressions above, it can be easily verified that

$$\mathbf{w}_- \odot \nabla_{w_+} L + \mathbf{w}_+ \odot \nabla_{w_-} L = 0, \tag{57}$$

and we will frequently use this relation later. Recall the definition of $\kappa_j = \mathbf{w}_{+;j} \mathbf{w}_{-;j}$, we now present useful lemmas before proving Theorem 4.1.

**Lemma B.1.** *Let* $\kappa_j(t) = \mathbf{w}_{+;j}(t) \mathbf{w}_{-;j}(t)$, $\gamma_{k;j}^{\mathbf{w}\pm}$ *denote the $j$-th component of* $\gamma_k^{\mathbf{w}\pm}$, *and*

$$\epsilon_j(t) = \int_0^t ds \left( \frac{\gamma_{k;j}^{\mathbf{w}_+}(s)}{\mathbf{w}_{+;j}(s)} + \frac{\gamma_{k;j}^{\mathbf{w}_-}(s)}{\mathbf{w}_{-;j}(s)} \right), \tag{58}$$

*then we have*

$$\kappa_j(t) = \kappa_j(0) e^{-\eta \epsilon_j(t)}. \tag{59}$$

*Proof.* The proof applies the dynamics of $\mathbf{w}_+$ and that of $\mathbf{w}_-$:

$$\frac{d\kappa}{dt} = \dot{\mathbf{w}}_+ \odot \mathbf{w}_- + \dot{\mathbf{w}}_- \odot \mathbf{w}_+$$

$$= \left( -\frac{\nabla_{\mathbf{w}_+} L}{1 - \mu} - \eta \gamma_k^{\mathbf{w}_+} \right) \odot \mathbf{w}_- + \mathbf{w}_+ \odot \left( -\frac{\nabla_{\mathbf{w}_-} L}{1 - \mu} - \eta \gamma_k^{\mathbf{w}_-} \right)$$

$$= -\eta \left( \gamma_k^{\mathbf{w}_+} \odot \mathbf{w}_- + \mathbf{w}_+ \odot \gamma_k^{\mathbf{w}_-} \right), \tag{60}$$

where we use Eq. (12) in the second equality and Eq. (57) in the third equality. As a result, for the $j$-th component of $\kappa$, we have

$$\dot{\kappa}_j = -\eta \kappa_j \left( \frac{\gamma_{k;j}^{\mathbf{w}_+}}{\mathbf{w}_{+;j}} + \frac{\gamma_{k;j}^{\mathbf{w}_-}}{\mathbf{w}_{-;j}} \right)$$

$$\implies \kappa_j(t) = \kappa_j(0) e^{-\eta \epsilon_j(t)}. \tag{61}$$

$\square$

It is also interesting to investigate the dynamics of $\mathbf{w}$ as shown below.

**Lemma B.2.** *If $\mathbf{w}_\pm$ is run with HBF, then the dynamics of $\mathbf{w}$ satisfies that*

$$\dot{\mathbf{w}} = -4\mathbf{v} \odot \frac{\nabla_{\mathbf{w}} L}{1 - \mu} - \eta \Gamma_k^{\mathbf{w}} \tag{62}$$

*where we let*

$$\mathbf{v} = (\mathbf{w}_+ \odot \mathbf{w}_+ + \mathbf{w}_- \odot \mathbf{w}_-), \quad \Gamma_k^{\mathbf{w}} = 2 \left( \gamma_k^{\mathbf{w}_+} \odot \mathbf{w}_+ - \gamma_k^{\mathbf{w}_-} \odot \mathbf{w}_- \right). \tag{63}$$

*Proof.* Using the dynamics of $\mathbf{w}_\pm$ Eq. (12), we can show that

$$\dot{\mathbf{w}} = 2\dot{\mathbf{w}}_+ \odot \mathbf{w}_+ - 2\dot{\mathbf{w}}_- \odot \mathbf{w}_-$$

$$= 2 \left( -\frac{\nabla_{\mathbf{w}_+} L}{1 - \mu} - \eta \gamma_k^{\mathbf{w}_+} \right) \odot \mathbf{w}_+ - 2 \left( -\frac{\nabla_{\mathbf{w}_-} L}{1 - \mu} - \eta \gamma_k^{\mathbf{w}_-} \right) \odot \mathbf{w}_-$$

$$= -4 \left( \mathbf{w}_+ \odot \mathbf{w}_+ + \mathbf{w}_- \odot \mathbf{w}_- \right) \odot \frac{\nabla_{\mathbf{w}} L}{1 - \mu} - 2\eta \left( \gamma_k^{\mathbf{w}_+} \odot \mathbf{w}_+ - \gamma_k^{\mathbf{w}_-} \odot \mathbf{w}_- \right). \tag{64}$$

$\square$

To show the implicit bias of HBF, we need to first explore the dynamics of $\mathbf{w}$, which is present in the following lemma.

**Lemma B.3** (Dynamics of $\mathbf{w}$ for diagonal linear networks under HBF). *Under conditions of Theorem 4.1, if the diagonal linear network $f(x; \mathbf{w})$ is trained with HBF (Theorem 3.1), let*

$$\Lambda_j^{\mathrm{GF}}(\mathbf{w}; \kappa(t)) = \frac{2\kappa_j(t)}{4} \left[ \frac{\mathbf{w}_j(t)}{2\kappa_j(t)} \operatorname{arcsinh}\left( \frac{\mathbf{w}_j(t)}{2\kappa_j(t)} \right) - \sqrt{1 + \frac{\mathbf{w}_j^2(t)}{4\kappa_j^2(t)}} + 1 \right]$$

$$\varphi_j(t) = \frac{\eta}{4} \int_0^t ds \left[ \frac{\gamma_{k;j}^{\mathbf{w}_+}(s)}{\mathbf{w}_{+;j}(s)} - \frac{\gamma_{k;j}^{\mathbf{w}_-}(s)}{\mathbf{w}_{-;j}(s)} \right]$$

$$\Lambda_j(\mathbf{w}, t; \kappa) = \Lambda_j^{\mathrm{GF}}(\mathbf{w}; \kappa(t)) + \mathbf{w}_j(t) \varphi_j(t), \tag{65}$$

*then the learning dynamics of the parameter $\mathbf{w}$ satisfies that*

$$\frac{d}{dt} \partial_{\mathbf{w}_j} \Lambda_j + \frac{\partial_{\mathbf{w}_j} L}{1 - \mu} = 0. \tag{66}$$

The proof of this lemma can be found in Appendix B.2. In the following we first focus on the proof of Theorem 4.1.

### B.1   PROOF OF THEOREM 4.1

Now we can prove Theorem 4.1 with above helper lemmas.

*Proof.* Recall the definition of $\Lambda_j$ in Lemma B.3 and we further define

$$\Lambda(\mathbf{w}, t; \kappa) = \sum_{j=1}^{d} \Lambda_j(\mathbf{w}, t; \kappa), \tag{67}$$

then Lemma B.3 gives us

$$\frac{d}{dt}\nabla_{\mathbf{w}}\Lambda(\mathbf{w},t;\kappa) = \left(\frac{d}{dt}\partial_{\mathbf{w}_1}\Lambda_1(\mathbf{w},t;\kappa),\ldots,\frac{d}{dt}\partial_{\mathbf{w}_d}\Lambda_d(\mathbf{w},t;\kappa)\right)^T$$

$$= -\frac{X^T r}{n(1-\mu)}$$

$$\implies \nabla_{\mathbf{w}}\Lambda(\mathbf{w}(\infty),\infty;\kappa(\infty)) - \nabla_{\mathbf{w}}\Lambda(\mathbf{w}(0),0;\kappa(0)) = -\sum_{i=1}^{n}\frac{x_i\int_0^\infty r_i(\tau)d\tau}{n(1-\mu)} = \sum_{i=1}^{n}x_i c_i \quad (68)$$

where we let $c_i = -\frac{\int_0^\infty r_i(\tau)d\tau}{n(1-\mu)}$. Let $\nabla_{\mathbf{w}}\Lambda(\mathbf{w}(0),0;\kappa(0)) = 0$ and recall the definition of $\Lambda(\mathbf{w};\kappa)$ in Theorem 4.1., then Eq. (68) is equivalent to

$$\nabla_{\mathbf{w}}\Lambda(\mathbf{w};\kappa) - \sum_{i=1}^{n}x_i c_i = 0,$$

which is exactly the KKT condition of $\arg\min_{\mathbf{w}:X\mathbf{w}=y}\Lambda(\mathbf{w};\kappa)$ proposed in Theorem 4.1. Therefore, we finish the proof. $\qquad\square$

### B.2 PROOF OF LEMMA B.3

In this section we present the proof of Lemma B.3.

*Proof.* For simplicity, in the following we write the subscripts explicitly. According to Lemma B.2, the dynamics of $\mathbf{w}_j$ can be written as

$$\dot{\mathbf{w}}_j = -\frac{4}{1-\mu}\mathbf{v}_j\partial_{\mathbf{w}_j}L - \eta\Gamma_{k;j}^{\mathbf{w}}. \quad (69)$$

Note that

$$\mathbf{v}_j^2 - \mathbf{w}_j^2 = 4\mathbf{w}_{+;j}^2\mathbf{w}_{-;j}^2 \implies \mathbf{v}_j^2 = \sqrt{\mathbf{w}_j^2 + 4\kappa_j^2}, \quad (70)$$

then Eq. (69) can be written as

$$\frac{\dot{\mathbf{w}}_j}{4\sqrt{\mathbf{w}_j^2 + 4\kappa_j^2}} = -\frac{\partial_{\mathbf{w}_j}L}{1-\mu} - \eta\frac{\Gamma_{k;j}^{\mathbf{w}}}{4\sqrt{\mathbf{w}_j^2 + 4\kappa_j^2}}. \quad (71)$$

We now define a function

$$\Lambda_j(\mathbf{w},t;\kappa) = \bar{\Lambda}_j(\mathbf{w},t;\kappa) + \mathbf{w}_j\varphi_j(t) \quad (72)$$

for some $\bar{\Lambda}_j(\mathbf{w},t;\kappa)$ and $\varphi_j(t)$ such that

$$\frac{d}{dt}\partial_{\mathbf{w}_j}\Lambda_j(\mathbf{w},t;\kappa) = \frac{\dot{\mathbf{w}}_j + \eta\Gamma_{k;j}^{\mathbf{w}}}{4\sqrt{\mathbf{w}_j^2 + 4\kappa_j^2}}, \quad (73)$$

the we can prove this lemma. Now we continue to find the $\bar{\Lambda}_j(\mathbf{w},t;\kappa)$ and $\varphi_j(t)$. By definition,

$$\frac{d}{dt}\partial_{\mathbf{w}_j}\Lambda_j(\mathbf{w},t;\kappa) = \partial_{\mathbf{w}_j}^2\bar{\Lambda}_j\dot{\mathbf{w}}_j + \partial_t\partial_{\mathbf{w}_j}\bar{\Lambda}_j + \dot{\varphi}_j, \quad (74)$$

which, when compared with Eq. (73), implies that

$$\partial_{\mathbf{w}_j}^2\bar{\Lambda}_j = \frac{1}{4\sqrt{\mathbf{w}_j^2 + 4\kappa_j^2}}. \quad (75)$$

Solving this equation gives us

$$\partial_{\mathbf{w}_j}\bar{\Lambda}_j = \frac{1}{4}\int\frac{d\mathbf{w}_j}{\sqrt{\mathbf{w}_j^2 + 4\kappa_j^2}} = \frac{\ln\left(\sqrt{\mathbf{w}_j^2 + 4\kappa_j^2} + \mathbf{w}_j\right)}{4} + c \quad (76)$$

where $c$ is a constant and can be determined by requiring $\partial_{\mathbf{w}_j}\bar{\Lambda}_j|_{t=0} + \varphi_j(0) = 0 \implies c = -\ln(2\kappa_j(0))/4$. Thus Eq. (76) becomes

$$\partial_{\mathbf{w}_j}\bar{\Lambda}_j = \frac{1}{4}\ln\left(\frac{\sqrt{\mathbf{w}_j^2 + 4\kappa_j^2(t)} + \mathbf{w}_j}{2\kappa_j(t)}\right) - \frac{\eta\epsilon_j(t)}{4}$$

where we have used the definition of $\epsilon_j(t)$ in Lemma B.1. Solving the above equation will give us the form of $\bar{\Lambda}_j$

$$\begin{aligned}
\bar{\Lambda}_j &= \frac{1}{4}\int d\mathbf{w}_j \operatorname{arcsinh}\left(\frac{\mathbf{w}_j}{2\kappa_j(t)}\right) - \frac{\eta\epsilon_j(t)\mathbf{w}_j}{4} \\
&= \frac{1}{4}\left[\mathbf{w}_j\operatorname{arcsinh}\left(\frac{\mathbf{w}_j}{2\kappa_j(t)}\right) - \sqrt{\mathbf{w}_j^2 + 4\kappa_j^2(t)} + 2\kappa_j(t)\right] - \frac{\eta\epsilon_j(t)\mathbf{w}_j}{4} \\
&= \Lambda_j^{\mathrm{GF}}(\mathbf{w};\kappa(t)) - \frac{\eta\epsilon_j(t)\mathbf{w}_j}{4}
\end{aligned} \tag{77}$$

where we use the definition of $\Lambda^{\mathrm{GF}}$ in Eq. (11). Comparing the rest parts of Eq. (74) with Eq. (73) requires that

$$\partial_t\partial_{\mathbf{w}_j}\bar{\Lambda}_j + \dot{\varphi}_j = \eta\frac{\Gamma_{k;j}^{\mathbf{w}}}{4\sqrt{\mathbf{w}_j^2 + 4\kappa_j^2(t)}}$$

$$\implies \dot{\varphi}_j(t) = \frac{\eta\kappa_j^2(t)\dot{\epsilon}_j(t)}{\left(\mathbf{w}_j + \sqrt{\mathbf{w}_j^2 + 4\kappa_j^2(t)}\right)\sqrt{\mathbf{w}_j^2 + 4\kappa_j^2(t)}} + \eta\frac{\Gamma_{k;j}^{\mathbf{w}}}{4\sqrt{\mathbf{w}_j^2 + 4\kappa_j^2(t)}}. \tag{78}$$

When combined with the form of $\bar{\Lambda}_j$, we can find the form of $\Lambda_j$:

$$\begin{aligned}
\Lambda_j(\mathbf{w},t;\kappa) &= \Lambda_j^{\mathrm{GF}}(\mathbf{w};\kappa(t)) + \eta\mathbf{w}_j\int\frac{ds}{\sqrt{\mathbf{w}_j^2 + 4\kappa_j^2(s)}}\left[\frac{\kappa_j^2(s)}{\mathbf{w}_j + \sqrt{\mathbf{w}_j^2 + 4\kappa_j^2(s)}}\dot{\epsilon}_j \right. \\
&\qquad\left. - \frac{\sqrt{\mathbf{w}_j^2 + 4\kappa_j^2(s)}\dot{\epsilon}_j}{4} + \frac{\Gamma_{k;j}^{\mathbf{w}}}{4}\right] \\
&= \Lambda_j^{\mathrm{GF}}(\mathbf{w};\kappa(t)) + \eta\mathbf{w}_j\int\frac{ds}{\sqrt{\mathbf{w}_j^2 + 4\kappa_j^2(s)}}\left[-\frac{\mathbf{w}_j\dot{\epsilon}_j}{4} + \frac{\Gamma_{k;j}^{\mathbf{w}}}{4}\right] \\
&= \Lambda_j^{\mathrm{GF}}(\mathbf{w};\kappa(t)) + \eta\mathbf{w}_j\int ds\left[\frac{\gamma_{k;j}^{\mathbf{w}_+}}{\mathbf{w}_{+;j}} - \frac{\gamma_{k;j}^{\mathbf{w}_-}}{\mathbf{w}_{-;j}}\right]
\end{aligned} \tag{79}$$

where we use the definition of $\epsilon_j$ (Lemma B.1) and Eq. (70) in the last equality. $\square$

## B.3 IMPLICIT BIAS OF HBF FOR DIAGONAL LINEAR NETWORKS WHEN $\alpha = 2$

In this case, the correction term $\gamma^{\mathbf{w}\pm}$ will be

$$\gamma^{\mathbf{w}\pm} = \frac{1+\mu}{2(1-\mu)^3}\nabla_{\mathbf{w}_\pm}L \cdot \nabla_{\mathbf{w}_\pm}^2 L.$$

We need to first find the Hessian $\nabla_{\mathbf{w}_\pm}^2 L$. Due to the element-wise product, it will be convenient to derive the Hessian by writing the subscripts explicitly. We start with $\mathbf{w}_+$.

$$\begin{aligned}
\partial_{\mathbf{w}_{+;i}}\partial_{\mathbf{w}_{+;j}}L &= \frac{2}{n}\partial_{\mathbf{w}_{+;i}}\left(\mathbf{w}_{+;j}(X^Tr)_j\right) \\
&= \frac{2}{n}\left[\delta_{ij}(X^Tr)_j + \sum_{c=1}^n\mathbf{w}_{+;j}\partial_{\mathbf{w}_{+;i}}\left(x_{c;j}(x_c^T\mathbf{w} - y_c)\right)\right] \\
&= \frac{2}{n}\left[\delta_{ij}(X^Tr)_j + 2\sum_{c=1}^n\mathbf{w}_{+;j}x_{c;j}x_{c;i}\mathbf{w}_{+;i}\right],
\end{aligned} \tag{80}$$

where we use the delta symbol $\delta_{ij} = 1$ if $i = j$ otherwise $\delta_{ij} = 0$. Therefore, we can conclude that

$$\nabla^2_{\mathbf{w}_+} L = \frac{2}{n} \left[ \text{diag}(X^T r) + 2 \sum_{c=1}^n (\mathbf{w}_+ \odot x_c)(\mathbf{w}_+ \odot x_c)^T \right]. \tag{81}$$

Following a similar approach, we obtain that for $\mathbf{w}_-$

$$\partial_{\mathbf{w}_{-;i}} \partial_{\mathbf{w}_{-;j}} L = -\frac{2}{n} \partial_{\mathbf{w}_{-;i}} \left( \mathbf{w}_{-;j}(X^T r)_j \right)$$

$$= \frac{2}{n} \left[ -\delta_{ij}(X^T r)_j + 2 \sum_{c=1}^n \mathbf{w}_{-;j} x_{c;j} x_{c;i} \mathbf{w}_{-;i} \right] \tag{82}$$

$$\implies \nabla^2_{\mathbf{w}_-} L = \frac{2}{n} \left[ -\text{diag}(X^T r) + 2 \sum_{c=1}^n (\mathbf{w}_- \odot x_c)(\mathbf{w}_- \odot x_c)^T \right]. \tag{83}$$

It is now left for us to find the form of $\nabla_{\mathbf{w}_\pm} L \cdot \nabla^2_{\mathbf{w}} L$. Again, it is convenient to write the subscripts explicitly:

$$\left( \nabla_{\mathbf{w}_+} L \cdot \nabla^2_{\mathbf{w}_+} L \right)_j = \sum_{i=1}^d \partial_{\mathbf{w}_{+;i}} \partial_{\mathbf{w}_{+;j}} L \partial_{\mathbf{w}_{+;i}} L$$

$$= \frac{4}{n^2} \sum_{i=1}^d \left[ \delta_{ij}(X^T r)_j + 2 \sum_{c=1}^n \mathbf{w}_{+;j} x_{c;j} x_{c;i} \mathbf{w}_{+;i} \right] \mathbf{w}_{+;i}(X^T r)_i$$

$$= \frac{4}{n^2} \left[ \mathbf{w}_{+;j}((X^T r)_j)^2 + 2 \sum_{c=1}^n \mathbf{w}_{+;j} x_{c;j} \left( x_c \odot \mathbf{w}_+ \odot \mathbf{w}_+ \right)^T X^T r \right]. \tag{84}$$

Similarly,

$$\left( \nabla_{\mathbf{w}_-} L \cdot \nabla^2_{\mathbf{w}_-} L \right)_j = \frac{4}{n^2} \left[ \mathbf{w}_{-;j}((X^T r)_j)^2 - 2 \sum_{c=1}^n \mathbf{w}_{-;j} x_{c;j} \left( x_c \odot \mathbf{w}_- \odot \mathbf{w}_- \right)^T X^T r \right]. \tag{85}$$

Using Eq. (84) and (85), we can derive that

$$\frac{\gamma_j^{\mathbf{w}_\pm}}{\mathbf{w}_{\pm;j}} = \frac{2(1+\mu)}{(1-\mu)^3 n^2} \left[ ((X^T r)_j)^2 \pm 2 \sum_{c=1}^n x_{c;j} \left( x_c \odot \mathbf{w}_\pm \odot \mathbf{w}_\pm \right)^T X^T r \right], \tag{86}$$

which further gives us the integral $\epsilon_j$:

$$\dot{\epsilon}_j = \frac{\gamma_j^{\mathbf{w}_+}}{\mathbf{w}_{+;j}} + \frac{\gamma_j^{\mathbf{w}_-}}{\mathbf{w}_{-;j}}$$

$$= \frac{4(1+\mu)}{(1-\mu)^3 n^2} \left[ ((X^T r)_j)^2 + \sum_{c=1}^n \sum_{i=1}^d x_{c;j} x_{c;i}(X^T r)_i \left( \mathbf{w}^2_{+;i} - \mathbf{w}^2_{-;i} \right) \right]$$

$$= \frac{4(1+\mu)}{(1-\mu)^3 n^2} \left[ ((X^T r)_j)^2 + \sum_{c=1}^n x_{c;j} x_c^T (\mathbf{w} \odot (X^T r)) \right]$$

$$= \frac{4(1+\mu)}{(1-\mu)^3} \left[ (\nabla_{\mathbf{w}} L)_j^2 + \frac{1}{n} \left( X^T X(\mathbf{w} \odot \nabla_{\mathbf{w}} L) \right)_j \right]. \tag{87}$$

On the other hand, according to Lemma B.2, $\partial_{\mathbf{w}_i} L$ can be written as

$$-(1-\mu) \frac{\dot{\mathbf{w}}_i}{4\mathbf{v}_i} - \eta(1-\mu) \frac{\Gamma_i^{\mathbf{w}}}{4\mathbf{v}_i}, \tag{88}$$

which further gives us that

$$\eta \int_0^t ds \left( X^T X (\mathbf{w} \odot \nabla_{\mathbf{w}} L) \right)_j = -\eta(1-\mu) \sum_{c=1}^n \sum_{i=1}^d x_{c;j} x_{c;i} \int_{\mathbf{w}_i(0)}^{\mathbf{w}_i(t)} d\mathbf{w}_i \frac{\mathbf{w}_i(s)}{4\mathbf{v}_i(s)} + \mathcal{O}\left(\eta^2\right)$$

$$= -\eta(1-\mu) \sum_{c=1}^n \sum_{i=1}^d x_{c;j} x_{c;i} \int_{\mathbf{w}_i(0)}^{\mathbf{w}_i(t)} d\mathbf{w}_i \frac{\mathbf{w}_i(s)}{4\sqrt{\mathbf{w}_i^2(s) + 4\kappa_i^2(s)}} + \mathcal{O}\left(\eta^2\right)$$

$$= -\frac{\eta(1-\mu)}{4} \sum_{c=1}^n \sum_{i=1}^d x_{c;j} x_{c;i} \left( \sqrt{\mathbf{w}_i^2(t) + 4\kappa_i^2(t)} - \sqrt{\mathbf{w}_i^2(0) + 4\kappa_i^2(0)} \right).$$

where we use Lemma B.2 in the first equality and Eq. (70) in the second equality. Since $\mathbf{w}(0) = 0$ and Lemma B.1, we obtain

$$\eta \int_0^t ds \left( X^T X (\mathbf{w} \odot \nabla_{\mathbf{w}} L) \right)_j = -\frac{\eta(1-\mu)}{4} \sum_{c=1}^n \sum_{i=1}^d x_{c;j} x_{c;i} \left( \sqrt{\mathbf{w}_i^2(t) + 4\kappa_i^2(0)} - 2\kappa_i(0) \right)$$

$$= -\frac{\eta(1-\mu)}{4} \left( X^T X \mathbf{q}(t) \right)_j \tag{89}$$

where we let $\mathbf{q} \in \mathbb{R}^d$ with

$$\mathbf{q}_i(t) = \sqrt{\mathbf{w}_i^2(t) + 4\kappa_i^2(0)} - 2\kappa_i(0) \geq 0.$$

Now combining Eq. (87) and Eq. (89), we can derive

$$\eta \epsilon_j(t) = \frac{4\eta(1+\mu)}{(1-\mu)^3} \int_0^t ds (\partial_{\mathbf{w}_j} L)^2 - \frac{\eta(1+\mu)}{(1-\mu)^2 n} \left( X^T X \mathbf{q} \right)_j + \mathcal{O}\left(\eta^2\right). \tag{90}$$

To obtain the full potential function, we still need to find the form of $\varphi_j$. According to the definition of $\mathbf{v}$ and $\epsilon_j$ and Eq. (86), we can derive

$$2\gamma_{k;j}^{\mathbf{w}_+} \mathbf{w}_{+;j} - 2\gamma_{k;j}^{\mathbf{w}_-} \mathbf{w}_{-;j} - \mathbf{w}_j \dot{\epsilon}_j = \mathbf{v}_j \left( \frac{\gamma_{k;j}^{\mathbf{w}_+}}{\mathbf{w}_+} - \frac{\gamma_{k;j}^{\mathbf{w}_-}}{\mathbf{w}_-} \right)$$

$$= \frac{4(1+\mu)}{(1-\mu)^3 n} \sum_{c=1}^n \sum_{i=1}^d \mathbf{v}_j x_{c;j} x_{c;i} \mathbf{v}_i \partial_{\mathbf{w}_i} L, \tag{91}$$

which, when combined with the definition of $\varphi_j$ in Lemma B.3, further gives us

$$\dot{\varphi}_j = \eta \frac{(1+\mu)}{(1-\mu)^3 n} \sum_{c=1}^n \sum_{i=1}^d x_{c;j} x_{c;i} \mathbf{v}_i \partial_{\mathbf{w}_i} L$$

$$= -\frac{\eta(1+\mu)}{4(1-\mu)^2 n} \sum_{c=1}^n \sum_{i=1}^d x_{c;j} x_{c;i} \dot{\mathbf{w}}_i + \mathcal{O}\left(\eta^2\right) \tag{92}$$

where we use Eq. (88) in the second equality. As a result,

$$\varphi_j(\infty) = -\frac{\eta(1+\mu)}{4(1-\mu)^2 n} \sum_{c=1}^n \sum_{i=1}^d x_{c;j} x_{c;i} \mathbf{w}_i(t) = \frac{\eta(1+\mu)}{4(1-\mu)^2 n} \left( X^T X \mathbf{w} \right)_j. \tag{93}$$

One interesting thing aspect of $\varphi_j$ if $\mathbf{w}$ converges to an interpolation solution where $X\mathbf{w}(\infty) = y$ is

$$\varphi_j(\infty) = \frac{\eta(1+\mu)}{4(1-\mu)^2} \partial_{\mathbf{w}_j} L(0). \tag{94}$$

In summary, the potential function $\mathcal{O}\left(\eta^2\right)$-close HBF is

$$\kappa_j(\infty) = \kappa_j(0) \exp \left( -\frac{4\eta(1+\mu)}{(1-\mu)^3} \int_0^\infty ds (\partial_{\mathbf{w}_j} L)^2 + \frac{\eta(1+\mu)}{(1-\mu)^2 n} \left( X^T X \mathbf{q}(\infty) \right)_j \right)$$

$$\Lambda_j(\mathbf{w}, \infty; \kappa) = \Lambda_j^{\mathrm{GF}}(\mathbf{w}, \kappa(\infty)) + \frac{\eta(1+\mu)}{4(1-\mu)^2} \mathbf{w}_j \partial_{\mathbf{w}_j} L(0). \tag{95}$$

## C   DETAILS FOR NUMERICAL EXPERIMENTS

### C.1   DETAILS FOR SECTION 5

For the discrete learning dynamics of HB and GD, we set the learning rate as $\eta$ and the momentum factor is $\mu$. For the continuous approximations, we use $\eta_{\text{Euler}} = \eta/10$ as the Euler step sizes to approximate the dynamics. These hyper-parameters are listed in Table 2. We let the model parameter

| $x, y$ | $1, 0.6$ |
|---|---|
| Starting point | $a_1 = 2.8, a_2 = 3.5$ |
| $\eta$ | $5 \times 10^{-3}$ |
| $\mu$ | $0.7$ |
| $\eta_{\text{Euler}}$ | $5 \times 10^{-4}$ |

Table 2: hyper-parameters for 2d model

be $\beta = (a_1, a_2)^T \in \mathbb{R}^2$. For RGF, we use the ODE $\dot{\beta} = -\frac{\nabla_\beta L}{1-\mu} \implies \beta_{k+1} = \beta_k - \eta_{\text{Euler}} \frac{\nabla_\beta L}{1-\mu}$. Formulations of HBFs with $\alpha = 2, 3$ are denoted in Table 1.

### C.2   DETAILS FOR SECTION 5

We denote $\mathbf{1}_d = (1, \ldots, 1)^T \in \mathbb{R}^d$. For the dataset $\{(x_i, y_i)\}_{i=1}^d$, we set $n = 40, d = 100$. The data point follows a Gaussian distribution $\mathcal{N}(0, I_d)$. To make the ground truth solution $\mathbf{w}^*$ sparse, we let 5 components of it be nonzero. Recall that the initialization is $\kappa(0) = s^2 \mathbf{1}_d$ where $s$ controls the initialization scale. In Fig. 2(a), we make the initialization as $\mathbf{w}_+ = \mathbf{w}_- = s\mathbf{1}_d$ with $s = 0.01$. We set the learning rate $\eta$ for HB as $10^{-3}$. For RGF and HBF, we let the Euler step size $\eta_{\text{Euler}} = 10^{-4}$ to simulate the continuous dynamics. In Fig. 2(b) and 2(c), we set $\eta = 10^{-2}$. For the initialization, to make the task slightly harder, we let $\mathbf{w}_+ = \vartheta s \mathbf{1}_d$ and $\mathbf{w}_- = s\mathbf{1}_d/\vartheta$ with $\vartheta = 0.9$ such that we still have $\kappa(0) = s^2 \mathbf{1}_d$ while the initialization symmetry is slightly broken.

