# OpenReview forum: "Continuous Approximation of Momentum Methods with Explicit Discretization Error"
_ICLR.cc/2025/Conference — ICLR 2025 Conference Withdrawn Submission_

### Official Review · Reviewer_7uEH · 2024-10-25

**Soundness:** 2
**Presentation:** 2
**Contribution:** 2
**Rating:** 3
**Confidence:** 3

**Summary:**

This paper studies the relationship between discrete-type momentum methods (Heavy-Ball (HB); Algorithm (2)) and their continuous approximations (HB Flow (HBF); (3)) and shows that HBF is $\mathcal{O} (\eta^\alpha)$-close continuous approximation of HB (Theorem 3.1). Moreover, it studies the implicit bias of HBF (Theorem 4.1) for diagonal linear networks. In addition, it provides some numerical results to support their analyses.

**Strengths:**

The strengths of the paper are to investigate theoretically the gap between HB and HBF, to show that HBF is $\mathcal{O} (\eta^\alpha)$-close continuous approximation of HB (Theorem 3.1), and to provide the implicit bias of HBF (Theorem 4.1) for diagonal linear networks.

**Weaknesses:**

I have many concerns for the paper. Please see the following Questions. However, I might misread it. In that case, I apologize for my inconvenience.

**Questions:**

1. Definition 2.1 seems not to be defined explicitly. For example,  I cannot understand what "for a constant $C(T) > 0$ ..." is, since $T$ is not defined. Could you provide the explicit definition of an $\mathcal{O} (\eta^\alpha)$-close continuous approximation? (I guess that "for a constant $C(T) > 0$ ..." is replaced by "$\forall T \exists C(T) > 0$...").  I also guess that $\sup\_T C(T) < \infty$ is assumed, since HB seems to not close to HBF when $\limsup\_{T \to \infty} C(T) = \infty$. Anyway, we need mathematically explicit Definition 2.1.

2. Could you provide $C(T)$ in Theorem 3.1? I do not find that Appendix section (Proof of Theorem 3.1) includes the definition of $C(T)$ in Theorem 3.1. The explicit form of $C(T)$ is significant to check the gap between HB and HBF. Also, I cannot follow that  HBF is $\mathcal{O} (\eta^\alpha)$-close continuous approximation of HB, since Definition 2.1 is not well-defined.

3. $\gamma_k (\beta)$ in (3) or (5) is a function of $\alpha$. Can we replace  $\gamma_k (\beta)$ by e.g., $\gamma_{k,\alpha} (\beta)$?

4. Even if Theorem 3.1 is correct, I cannot understand why the theorem is important, that is, what the contribution of the paper is. Since I do not know the explicit form of $C(T)$ in Theorem 3.1, the following comments and questions are based on my unsupported claims:
- It is useful that the gap $\mathcal{O} (\eta^\alpha)$ should be small to train DNNs. Is it correct? Although I do not understand any significance and usefulness of Definition 2.1, the following are based on an objective such that the gap $\mathcal{O} (\eta^\alpha)$ should be small.
- If $C(T)$ is finite for any $T$ and if $\alpha \geq 1$ is fixed, then the learning rate $\eta  > 0$ should be small so that the gap $\mathcal{O} (\eta^\alpha)$ can be small. Is it correct?
- Let $\eta \in (0,1)$. Then, the parameter $\alpha  \geq 1$ should be large so that the gap $\mathcal{O} (\eta^\alpha)$ can be small. Is it correct? If it is correct, then why did the authors use only small $\alpha = 2, 3$ (see also Sections 3 and 4)?
- What are new/significant insights (e.g., appropriate setting of $\eta$, $\mu$, and $\alpha$ to train DNNs) obtained from Theorem 3.1?
- If $\limsup\_{T \to \infty} C(T) = \infty$ or $C(T)$ is very large for $T \geq t_0$ (where $t_0 \in \mathbb{N}$ is a certain step/time), then what is the finding obtained from Theorem 3.1? Remark: If it is theoretically guaranteed that $C(T) < \infty$ in Theorem 3.1 is small, then this question can be omitted.

5. Section 4 said there is a relationship between the bias of HBF and the generalization. I cannot understand the claim. The function $L$ is the empirical loss. Hence, HB and HBF using $L$ can be applied to only empirical risk minimization (ERM). Please explain what the connection between the bias of HBF (Theorem 4.1) based on ERM and generalization for expected risk minimization. As a result, I strongly believe that the authors can explain the relationship between Theorem 4.1 and the generalization. Moreover, what are new/significant insights (e.g., appropriate setting of $\eta$, $\mu$, and $\alpha$ to have better generalization) obtained from Theorem 4.1?

6. According to PyTorch (https://pytorch.org/docs/stable/generated/torch.optim.SGD.html#torch.optim.SGD), one practical HB is defined by (2) with a negative $\mu$ (torch.optim.SGD(params, lr=0.001, momentum=0.9, dampening=0, weight_decay=0, nesterov=False, *, maximize=False, foreach=None, differentiable=False, fused=None). Here, I have a question. What is HB used in the experiments? Just (2) with a positive $\mu$ such as: torch.optim.SGD(params, lr=0.001, momentum= - 0.9, dampening=0, weight_decay=0, nesterov=False, *, maximize=False, foreach=None, differentiable=False, fused=None) ? Please define HB and HBF used in the experiments explicitly.

---

### Official Review · Reviewer_3f8i · 2024-10-30

**Soundness:** 3
**Presentation:** 3
**Contribution:** 2
**Rating:** 5
**Confidence:** 4

**Summary:**

In this work, the authors propose a more accurate continuous approximation to discrete momentum updates which they call HBF that allows discretization error to higher arbitrary order. Then they use this HBF to study the implicit bias of HB on diagonal linear networks to understand the model's generalization property.

**Strengths:**

1) The paper extends the work of (https://openreview.net/forum?id=ZzdBhtEH9yB) and https://arxiv.org/abs/1906.04285  to find approximate continuous flows of higher order for momentum. This seems to be a nice contribution since higher order analysis for momentum methods has been missing in literature unlike for GD.
2) Table-1 shows that their derivation is consistent with previous works for momnetum and GD upto the first and second orders.

**Weaknesses:**

Although the authors show a promising analysis and derivation there are a couple of weakness that hinders the motivation and broad applicability of the work:

1) *Implications for higher order truncations*

a) The only contribution this paper has compared to it's past work such as https://openreview.net/forum?id=ZzdBhtEH9yB is the derivation for HBF $\alpha>2$ onwards. However, it is unclear about the significance of these higher order terms and how it changes the landscape trajectory. For example for $\alpha=2$, it has already been proved that the implicit regularization term is the gradient norm scaled by lr which drives trajectories to low sharp solutions. It can be quite easily shown that the gradient norm for the diagonal linear network loss is just the norm of the layers minimizing which is essentially minimizing the l1 norm of the solutions. The authors derive the same conclusion albiet through an effective initialization strategy. So, in my opinion Corollary 4.2 does not provide much new insights. Understanding the effect of $\alpha=3$ and so on would have been a significant contribution.

b) https://arxiv.org/abs/2302.01952 sheds light into instabilities in training and edge of stability by proposing modified flows that includes higher order terms. Is it possible to analyze such instabilities in training for HB. Does higher order term imply instabilities apart from just finding solutions that generalize well?

c) If we use modified GF with $\alpha=2$ but with a higher learning rate will that match the regularization effect for HBF with $\alpha=3$ in diagonal linear network? Although the authors made a distinction RGF and HBF, however it is unclear whether just using a large learning rate can match the regularization effect of HBF with $\alpha=3$. The advantage of momentum is not clear in this case.

2) *Validity of higher order approximation: upper bound on learning rate*

All the higher order truncations implicitly assumes that the learning rate and momentum is small. But there is no quantitative analysis on how small these quantities need to be for the $\alpha=2$ and $\alpha=3$ to hold.

3) *Limitation of model assumption*

Diagonal linear networks are far from being ideal deep network models. Although the loss is nonconvex, all the global minima are connected. This architecture misses the case where momentum is most effective, escaping local minimas and isolated basins. So, it is doubtful that such simple models can truly capture the advantages momentum can have apart from just the advantages GD posses.

**Questions:**

Each of the points raised in weakness-1 raises a question. In line-261, the authors mention that their derivation strategy is different than that of https://openreview.net/forum?id=ZzdBhtEH9yB. Can the authors specify how is it different, the proof strategy appears to be exactly the same.

---

### Official Review · Reviewer_P2X1 · 2024-11-03

**Soundness:** 3
**Presentation:** 3
**Contribution:** 3
**Rating:** 6
**Confidence:** 3

**Summary:**

Inspired by Miyagawa (2022), this paper proposes **Heavy Ball Flow (HBF)**, a continuous first-order ODE that can approximate the discrete HB with *arbitrary precision*. This is then applied to derive new implicit bias results for diagonal linear networks distinct from gradient flow. The theoretical claims are verified over a synthetic 2d example and two-layer diagonal linear networks.

**Strengths:**

- Well-written and easy to follow.
- Considers a relatively understudied problem and provides a solid, new approach to tackling HB from a continuous perspective in a numerically accurate manner (arbitrary accuracy)
- The newly derived implicit bias of HB in diagonal linear networks is quite interesting and surprisingly intuitive.

**Weaknesses:**

- Discussions on the implicit bias of HB for diagonal linear networks should be much more elaborated; notably, the relation to prior work [1] is completely missing. By considering a second-order ODE approximation to HB, [1] showed several theoretical results that are qualitatively similar to the implicit biases shown by the authors.
    - [1] showed that the quantity $\lambda := \frac{\eta}{(1 - \beta)^2}$ plays an important role in the implicit bias. This is similar to the quantity $\frac{\eta (1 + \mu)}{(1 - \mu)^2}$ that the authors claimed was crucial in understanding how HB enjoys better sparsity than GD.
    - [1] showed that the implicit regularization minimizes some Bregman divergence with potential induced by the "asymptotic balancedness" of the solution between the optimal solution and some perturbed initialization. This is similar to the authors' Theorem 4.1, where HBF is shown to effectively rescale the initialization and break the symmetry of the initialization.

- (continuing) If there are some differences, the authors should highlight which parts of [1] are loose compared to the new implicit bias results. If there are similarities (e.g., they are equivalent in some sense) or some parts that elucidate the connection of the new results to the known results of [1], those should also be highlighted.

- The authors should try to give a bit more detailed proof sketch of Theorem 3.1, highlighting the difficulty of extending the prior proof technique of Miyagawa (2022).

- [minor] typo in Eqn. (35) in the Appendix.

[1] https://proceedings.mlr.press/v238/papazov24a.html

**Questions:**

- Can the authors elaborate on whether such arbitrary precision continuous approximation is possibly only for first-order ODE? Indeed, as the authors have discussed in the Introduction, second-order ODE approximations for HB have been extensively studied, with their own benefits. For instance, Kovachki and Stuart (2022) stated that second-order (Hamiltonian) ODE can better capture the intuitions of HB's transient regime than first-order ODE approximation.
  - Could this be a further solution? Defining the velocity component $v_t = \dot{\beta_t}$, one could consider an autonomous first-order ODE of the form $(\dot{v_t}, \dot{\beta_t}) = (- v_t - \nabla F(\beta_t), v_t) + \text{correction term}$. Could this give more benefits than the currently proposed HBF?

- I believe the authors have missed this prior work: https://ojs.aaai.org/index.php/AAAI/article/view/26209, which also considers a second-order ODE approximation of HB.

- [low priority] As in the above AAAI'23 paper, it would be interesting to see if HB leads to new implicit bias results in overparametrized linear regression, even just empirically.

---

### Official Review · Reviewer_vbP4 · 2024-11-04

**Soundness:** 3
**Presentation:** 2
**Contribution:** 2
**Rating:** 5
**Confidence:** 3

**Summary:**

This work proposes a continuous time framework--HB Flow (HBF)--for the heavy-ball momentum method (HB). The authors show HBF approximate HB up to arbitrary order. They also study the implicit bias of HBF on diagonal linear networks which brings insights into the implicit bias problem in momentum methods. Experiments on simple neural networks are done to verify their results.

**Strengths:**

1. The authors propose HB flow--a continuous-time approximation of the heavy ball momentum method. By adding back the discretization error in each time interval, the HB flow achieves an arbitrarily close distance to the discrete-time dynamic. This is justified rigorously in theorem 3.1

2. More precisely, the distance between two dynamics is measured using $L^\infty[0, T]$ on the trajectory space. The $L^\infty$ is qualified by an upper bound in the learning rate $\eta^\alpha$. The authors give the specific form for the case $\alpha=1,2,3$ as examples, which help the readers to understand the construction of the continuous dynamic.

3. The authors computed the implicit bias of HBF for diagonal linear networks in theorem 4.1 which makes this framework useful in studying the generalization properties of HB. This type of behavior cannot be captured by lower order models like RGF.

4. In numerical experiments, Figure 1 illustrate the trajectories in a clear way and verifies main results from the paper.

**Weaknesses:**

1. I found the main theorem hard to follow. Specifically, Theorem 3.1 is stated with many notations without enough intuition/explanations. I am trying to understand this at a high level. So basically find the difference between continuous and discrete dynamics and use Taylor expansion for it. The order $\alpha$ in $\eta^\alpha$ just means how many times we do Taylor expansion in $t$ because the time interval is of length $\eta$. Although the computation needed to find the exact expressions is not trivial, one could argue this is not mathematically sophisticated.

2. Some notations are not consistently defined. For example, on page 4, the authors stated $G_k\in\mathbb{R}^d$, i.e. a d dimensional vector, but later $G_k$ becomes a function. Similar situation for $\gamma_k$.

**Questions:**

1. In Figure 1(a), I noticed that the $\alpha=2$ case gives a better approximation away from the convergence point while $\alpha=3$ is the opposite. I wonder if the authors have some intuition for this behavior.

2. This analysis can be generalized to the stochastic case by for example replacing $\nabla L$ by $\tilde{\nabla} L$ where $\tilde{\nabla}$ denotes the approximate gradient. The approximation can be done by either introducing a diffusion term [1] or an index switching process [2]. I think this might be an interesting future direction.

[1] Stochastic modified equations and dynamics of stochastic gradient algorithms I: Mathematical foundations, Li et al.

[2] Analysis of Stochastic Gradient Descent in Continuous Time, Latz.

---

### Note · Authors · 2024-12-04

**Comment:**

We thank all reviewers for the time and valuable comments. We will revise our manuscripts according to the constructive suggestions.

**Withdrawal Confirmation:**

I have read and agree with the venue's withdrawal policy on behalf of myself and my co-authors.